# Functional Thermoresponsive Hydrogel Molecule to Material Design for Biomedical Applications

**DOI:** 10.3390/polym14153126

**Published:** 2022-07-31

**Authors:** Sagar Pardeshi, Fouad Damiri, Mehrukh Zehravi, Rohit Joshi, Harshad Kapare, Mahendra Kumar Prajapati, Neha Munot, Mohammed Berrada, Prabhanjan S. Giram, Satish Rojekar, Faraat Ali, Md. Habibur Rahman, Hasi Rani Barai

**Affiliations:** 1Department of Pharmaceutical Technology, University Institute of Chemical Technology, KBC North Maharashtra University, Jalgaon 425001, Maharashtra, India; sagar.pardeshi201@gmail.com; 2Laboratory of Biomolecules and Organic Synthesis (BIOSYNTHO), Department of Chemistry, Faculty of Sciences Ben M’sick, University Hassan II of Casablanca, Casablanca 20000, Morocco; fouad.damiri@outlook.fr (F.D.); berrada_moh@hotmail.com (M.B.); 3Department of Clinical Pharmacy Girls Section, Prince Sattam Bin Abdul Aziz University Alkharj, Al-Kharj 11942, Saudi Arabia; mahrukh.zehravi@hotmail.com; 4Precision Nanosystems Inc., Vancouver, BC V6P 6T7, Canada; rohitjoshi07@gmail.com; 5Department of Pharmaceutics, Dr. D.Y. Patil Institute of Pharmaceutical Sciences and Research, Pune 41118, Maharashtra, India; hskapare@yahoo.in; 6Department of Pharmaceutics, School of Pharmacy and Technology Management, SVKM’s NMIMS, Shirpur 425405, Maharashtra, India; mahendraniper86@gmail.com; 7Department of Pharmaceutics, School of Pharmacy, Vishwakarma University, Pune 411048, Maharashtra, India; nehamunot@yahoo.com; 8Department of Pharmaceutical Sciences, University at Buffalo, The State University of New York, Buffalo, NY 14214, USA; 9Department of Pharmaceutical Sciences and Technology, Institute of Chemical Technology, Mumbai 400019, Maharashtra, India; 10Departments of Medicine and Pharmacological Sciences, Icahn School of Medicine at Mount Sinai, New York, NY 10029, USA; 11Laboratory Services, Department of Licensing and Enforcement, Botswana Medicines Regulatory Authority (BoMRA), Gaborone 999106, Botswana; frhtl6@gmail.com; 12Department of Global Medical Science, Wonju College of Medicine, Yonsei University, Wonju 26426, Korea; pharmacisthabib@yonsei.ac.kr; 13School of Mechanical and IT Engineering, Yeungnam University, Gyeongsan 38541, Korea

**Keywords:** hydrogel, thermoresponsive, release kinetics, biosensing, drug delivery

## Abstract

Temperature-induced, rapid changes in the viscosity and reproducible 3-D structure formation makes thermos-sensitive hydrogels an ideal delivery system to act as a cell scaffold or a drug reservoir. Moreover, the hydrogels’ minimum invasiveness, high biocompatibility, and facile elimination from the body have gathered a lot of attention from researchers. This review article attempts to present a complete picture of the exhaustive arena, including the synthesis, mechanism, and biomedical applications of thermosensitive hydrogels. A special section on intellectual property and marketed products tries to shed some light on the commercial potential of thermosensitive hydrogels.

## 1. Introduction

A hydrogel is a three-dimensional macromolecule network that is interlinked. It comprises hydrophilic or amphiphilic building blocks, which swell in water and may hold a lot of it. Cross linking makes hydrogels insoluble in water. To form a gel, water must account for at least 10% of the total volume or weight. Micro-sized water-soluble monomers, nanosize nanofibrils or nanotubes, and polymers make up hydrogels [1,2]. Crosslinking is a crucial characteristic that inhibits the gel from dissolving in water. Chemical and physical crosslinking are the two forms of crosslinking [3]. There is a covalent bond formed in chemical crosslinking, similar to the development of disulfide linkage, and this kind of gel would be permanent. Intermolecular forces, such as hydrogen bonding and other hydrophobic interactions or physical entanglements, are engaged in physical crosslinking [4]. Hydrogels have become the focus of considerable research in recent years. Their features, such as high-water content and the ability to control swelling characteristics, make them particularly appealing for biomedical applications [5]. In situ synthesized hydrogels can be used to create easy, “custom-made” treatments and diagnostics. Following the polymerization process, chemical or ionic crosslinking, or in reaction to an environmental stimulus such as temperature, pH, or ionic concentration of the external medium, a polymeric solution can be created and allowed to gel in situ [6].

Similarly, nanogels are nanosized hydrogels that combine the benefits of nanoscale materials with the benefits of hydrogels. Nanogels are distinguished by their small dimensions (up to 1000 nm) with high water absorption capacity. Thermoresponsive nanogels are delicate nanomaterials that react to variations in temperature. To make thermoresponsive nanogels, two methods have been used [7]. The primary structure of a nanogel-forming polymer is integrated with thermoresponsive polymer units in the first strategy to create thermoresponsiveness. To provide thermoresponsiveness, lipophilic moieties are bonded as a substituent to a hydrophilic polymer backbone in the second technique. The amount of interaction, which can be classified as hydrophobic or hydrophilic based on the change in free energy of the surrounding solvent, is the factor that causes thermoresponsiveness [8]. The hydrophobicity or hydrophilicity of a combination is indicated by a positive or negative variation in its free energy. Tailored drug delivery, the modulated release of drugs, imaging and tracking, and other applications of thermosensitive nanogels have all been investigated. Peters et al. developed a thermoresponsive PNIPMAM-based core-shell nanogel for the treatment of cancer that allows for regulated and triggered DOX release. With L929 fibroblasts, the toxicity of the produced nanogels was studied, which revealed low inherent cytotoxicity [8,9].

This review looks at hydrogels that react to changes in temperature. Their responsiveness to the thermal sensation is advantageous because heat is the single trigger for their gel formation, requiring no further chemical or environmental intervention. It can therefore be created, for example, when the temperature rises from atmospheric to physiological. Sol-gel shift is the term that describes the transformation from a solution to a gel [3,6,10]. Beyond a certain temperature, some hydrogels separate from the solution and solidify, and this threshold is defined as the lower critical solution temperature (LCST). The polymers are solvable below the LCST. They become progressively hydrophobic and insoluble above the LCST, resulting in gelation. Hydrogels generated by cooling a polymer solution, on the other hand, have an upper critical solution temperature (UCST). Spectroscopy, differential scanning calorimetry (DSC), and rheology can be used to confirm the sol-gel transformation of thermo-responsive hydrogels in the lab [11]. Due to their distinctive features, thermoresponsive hydrogels are extremely fascinating.

In the aqueous phase, poly(*N*-isopropylacrylamide) (NIPAAm) based polymers often had a very well defined lower critical solution temperature (LCST) of 32 °C, during which hydrogen bonds among water molecules and the NIPAAm chain form/break, accompanied by network hydration/dehydration. The swelling behavior of NIPAAm hydrogels varies dramatically over a small temperature range due to the delicate equilibrium between hydrophilic and hydrophobic nature [6,12]. Its temperature-modulated swelling feature has been used to create an on/off switch for various applications, including medication administration, controlled release, and so on. Introducing a hydrophobic or hydrophilic comonomer/crosslinker into the LCST could also be used to change the hydrophilic/hydrophobic equilibrium inside the polymer. Biomedical applications often demand the LCST between room temperature and body temperature (37 °C) and even higher [10,11]. This paper describes the uses of thermoresponsive gels in domains of interest to pharmaceutical and biological scientists and engineers in this context. It focuses on hydrogels made from natural, semi-synthetic, and synthetic polymers. Furthermore, this review illustrates phase transformation and release kinetics in several thermosensitive hydrogel systems. The paper describes the latest progress in thermoresponsive hydrogels for biological recognition, targeted drug delivery, tissue engineering, wound healing, and other applications [13,14,15].

## 2. Stimuli-Responsive Polymers

Temperature, ion concentration, redox processes, pH, light, shear stress, enzymes, and other triggers initiate stimuli-responsive polymers. The macroscopic behavior of polymer solutions is determined by their physical characteristics [16]. This even allows for some customization over the characteristics of biomedical devices and delivery systems comprised of such polymers, which has sparked a huge interest. The biological applicability of stimulus-responsive polymer systems allows only a limited range of stimuli, such as pH, ionic strength, and others, which limits their applicability. In this temperature range, various types of polymers exhibit thermosensitive characteristics [17].

### 2.1. Physicochemical Properties of Thermoresponsive Hydrogel

#### 2.1.1. Mechanical Strength

Thermosensitive hydrogels are delicate materials that alter physical characteristics in response to temperature changes, and they have a lot of applications in biomedicine [18,19]. Existing thermoresponsive hydrogels, such as poly(*N*-isopropylacrylamide) (PNIPAM), have poor mechanical properties and lack an energy-dissipating mechanism, making them unsuitable for potential implementation. Although some techniques, such as double networks, dual-cross-linking, and composites, can increase strength, they can diminish thermoresponsiveness. The development of thermosensitive hydrogels that retain inherent mechanical characteristics remains an important and pressing challenge. Hydrogel actuators with high strength, hardness, and stimuli sensitivity can be made by molecularly combining a hard element and a stimuli-sensitive component in a single polymer network [20]. Hydrogels that combine thermosensitive and hard compounds into a single system have recently been introduced by scientists at the University of California, resulting in high resilience and thermal responsive performance [21]. Thermal responsive PNIPAM was anchored onto the tough polymer backbone of poly (vinyl alcohol) (PVA) and its methacrylate derivatives using UV light irradiation (PVA-MA). The samples were dipped in a sodium sulphate salt solution after the one-pot polymerization to further enhance the hydrogel network, in which the PVA clumps and crystallizes under these circumstances. This hardening process does not decrease the stimuli-responsive qualities due to the single-network topology. The new hydrogel performed well mechanically and had a toughness (~10 MJ/m^3^) 100 times that of ordinary PNIPAM hydrogels [22].

#### 2.1.2. Adhesion

The development of robust and effective bonding among hydrogels and solid substances in wet environments, which would be critical for its assimilation into and efficiency in systems and devices, is still a continuous process. The excessive quantity of water in the polymeric matrix of hydrogels weakens their adhesive behaviour, as water generates a weak barrier that prevents actual surface contact between the hydrogels and substrates, resulting in reduced surface energy and decrease of adhesion strength [23]. Additionally, water molecules connect with the adhesive sites in the hydrogels via hydrogen bonding, reducing interfacial adhesion between the hydrogels and solid substances dramatically. If hydrogels are used in biomedicine, the situation becomes much more complex because most substrates (human bodies, tissues, and bioglues) are moist and delicate [18]. Researchers have investigated the chemical properties of marine animals, e.g., sandcastle worms and mytilus mussels, which can attach to a variety of immersed substrates in a hostile and unstable wet environment, to improve the adherence of a hydrogel to wet surfaces. Wet adhesives must meet numerous critical characteristics in comparison to dry adhesives, includes breaking down surface bound water layers, keeping cohesive against water corruption, and having a planned underwater healing [24]. Waite and Tanzer discovered a Catecholic amino acid (DOPA) in mussel foot plaque in 1981, and discovered that it contributes a lot to mussels’ powerful underwater adherence [25]. The catechol-containing peptide was found to be capable of penetrating aqueous boundary barriers and forming interfacial connections with underwater surfaces [20].

#### 2.1.3. Optical Property

Sensing, optoelectronics, nanomotion, theranostics, and biomedical applications are just a few of the uses for plasmonic compounds [26]. Plasmonic particles covered in thermosensitive polymers have plasmonic signals that are heat sensitive, which is of special importance. Poly-*N*-isopropyl acrylamide is commonly used in these systems (pNIPAM). At relatively low temperature, the polymer chains of pNIPAM are lengthened, and once the lower critical solution temperature (LCST) is achieved, the polymer experiences a hydrophilic-to-hydrophobic shift and collapses in aqueous phase [27]. As a result, in a gel comprising pNIPAM and plasmonic nanomaterials, the plasmonic signature changes due to either (1) an alteration in pairing in between nanomaterials as a feature of their detachment, (2) a modification in the refractive index of the polymer underlying the nanomaterials due to the removal of water from the polymer and substitution of the water with a greater refractive index organic polymer, or (3) both pathways [28]. Yang and colleagues demonstrated how these heat-sensitive substances might be employed, wherein silver nanoparticles were produced in situ within a polymeric gel consisting of pNIPAM [29]. Thermal responsive surface-enhanced Raman scattering (SERS) responses arise from a variation in polymer structure with heat. Feldman and colleagues demonstrated that optoelectronic actuators may be made using microgels of pNIPAM-coated gold nanorods. Other heat-responsive polymers could be used, as demonstrated by Liz-Marzan and colleagues [30]. They have used oligo (ethylene oxide) methacrylate as a heat responsive polymer conjugated from gold nanoparticles altered with bovine serum albumin whereby an atom transfer radical polymerization initiator was covalently attached [31,32].

##### Lasting Time

The lasting time or residence time of thermoresponsive hydrogels depends on various parameters, such as the nature and structure of the monomer, degree of cross-linking of the polymer, rate of biodegradation of polymer, or the intensity of the thermal stimulus. It plays an important role in deciding the fate of drug delivery. Plain thermosensitive polymers have poor mechanical strength and thus may get degraded in the biological fluid, rapidly limiting the sustained release of entrapped drugs. These polymers can be coupled with polymers having high mechanical strengths. Thus, the formed copolymers will have a greater lasting time as their degradation rate will be lowered, hence possibly leading to a prolonged residence time of the drug, ultimately improving the efficacy. Cui Z et al. studied the degradation property of copolymers synthesized using NIPAAm. It was found that at physiological conditions, LCST increased to approximately 35 °C from 37 °C after 20 days, indicating a slower rate of degradation and increased lasting time. It was also dependent on the mass loss property of the copolymer. Md Hasan Turabee et al., Synthesized *N*, *N*, *N*-trimethyl chitosan embedded thermosensitive Pluronic F127 hydrogel for the treatment of brain tumor. It was seen that, at pH 7.4, even after 30 days, the synthesized gel sustained the release of docetaxel for the treatment of glioma [33,34].

### 2.2. Natural Polymers and Their Derivatives

Many natural polymers and their derivates form a gel with temperature change. Thermosensitive polymers vary their solubility in response to changes in ambient temperature. Thermally responsive hydrogels can be made using these polymers alone or in combination with synthetic polymers [35].

#### 2.2.1. Polysaccharides

##### Cellulose Derivatives

Cellulose is insoluble in water. It was modified with methyl or hydroxypropyl groups to increase its water solubility. Methylcellulose (MC), a derivate of it, has been intensively studied for biomedical purposes. MC produces a thermoresponsive gel at 60–80 °C that cools to become a solution [36]. Methylcellulose was substituted with *N*-isopropyl acrylamide (NiPAAm), and the resulting polymer showed combined thermo-gelling characteristics of both polymers. Researchers also discovered that adding MC to NiPAAM polymers improves the hydrogel’s mechanical properties [37]. Stabenfeldt et al. functionalized methylcellulose with the laminin to fabricate a bioactive scaffold for neural tissue engineering. Methylcellulose was first oxidized (OXMC), followed by laminin tethering (OXMC+LN). OXMC+LN hydrogel enhanced neuronal cell adhesion and cell viability compared to MC and OXMC [38].

##### Chitosan

The deacetylation of chitin, which is found in the exoskeleton of crustaceans and insects, produces chitosan. Bhattarai et al. grafted poly (ethylene glycol) (PEG) into chitosan. PEG grafting improved the solubility of chitosan in water. Without a crosslinker, the PEG-grafted chitosan can form a thermoreversible hydrogel in physiological pH values [39].

Once evaluated with the model drug albumin, PEG-grafted chitosan exhibited controlled in-vitro drug release with an early burst and a continuous release for three days. When the PEG-grafted chitosan was crosslinked in situ using genipin, a low-cytotoxicity crosslinking agent, the quasi-linear release of drugs for up to 40 days was achievable. At 37 °C, though, the thermoreversible property of the hydrogel was lost [40]. A copolymer of chitosan of NiPAAm was prepared. The resultant hydrogel showed potential as cell carriers for tissue engineering applications and can be used to treat vesicoureteral reflux with minimal invasion [41]. Chitosan was conjugated with hydroxybutyl groups by Dang et al. [27]. At physiological temperatures, the formed hydrogel may gel in seconds and cool to a liquid state. When enclosed in hydrogels over two weeks, hMSC and cells originating from the intervertebral disc increased and produced an extracellular matrix. A chitosan-glycerophosphate salt (GP) hydrogel was tested for neural tissue engineering applications. Crompton et al. tested this hydrogel to polylysine-functionalized chitosan-GP to improve neuronal attachment and neurite development. More cells survived over nonmodified chitosan-GP at particular polylysine concentrations. This shows that polylysine-chitosan-GP could be a viable choice for brain tissue engineering [42].

##### Dextran

Dex-MA was created by crosslinking biodegradable dextran (Dex) with maleic anhydride (MA). Photo crosslinking the copolymer with NiPAAm made it more thermoresponsive. The resultant hydrogel was pH-sensitive and partly biodegradable [11]. Huang and colleagues used NiPAAm to copolymerize dextran oligolactate and 2-hydroxyethyl methacrylate (Dex-lactate-HEMA). The scientists determined that the drug release pattern is influenced by various parameters, including temperature, hydrogel swelling and degradation properties, and drug–hydrogel macromolecule interaction [43].

##### Xyloglucan

If more than 35% of the galactose remnants in xyloglucan are eliminated, it exhibits thermosensitive activity. Xyloglucan gels have been used to deliver drugs in a range of applications. Unfortunately, there is a lack of information on these hydrogels’ viscoelastic and morphological properties [13]. Nisbet et al. investigated the gelation characteristics and morphology of xyloglucan hydrogels under physiological circumstances. In comparison to deionized water, the existence of ions in PBS seems to affect the gelation. The optimum amount was discovered to be 3% (wt.) xyloglucan in aqueous media, which has a much greater elastic modulus than other natural or synthetic hydrogels. Furthermore, a gel might be freeze-dried at this concentration and analyzed using scanning electron microscopy. The images showed a macroporous, interconnected, three-dimensional network [44].

#### 2.2.2. Proteins

##### Gelatin

Gelatin is a thermoresponsive, biocompatible, and biodegradable polymer. The gelatin aqueous solution solidifies under 25 °C, while the gel reverts to fluid beyond 30 °C. Gelatin coupled with the other polymers demonstrates thermoresponsive gel formation near body temperature, which is desirable for biomedical applications [45]. Yang et al. created a gelatin hydrogel with monomethoxy poly (ethylene glycol)-poly (D, L-lactide) (mPEG-DLLA) block copolymers. At 37 °C, gelatin was combined with 30% wt. mPEG-DLLA and gelled quickly. Using gentamycin sulphate, hydrogels take five days or longer to release 50% of medication at ambient temperature. The whole duration of the release is 40 days. At 37 °C, the release profile is much slower.

Nevertheless, due to the deterioration of the hydrogel matrices, the release was no longer measurable after a week [46]. Ohya and Matsuda created a thermosensitive gel by grafting gelatin with NiPAAm. The resultant polymer displays a sol-gel transformation at physiologic temperatures and a pNiPAAm/gelatin (P/G) ratio larger than 5.8. Due to improved hydrophobicity and a favorable cell environment, a moderate hydrogel concentration (5% *w*/*v*) and a high P/G ratio resulted in higher cell growth and extracellular matrix formation [47]. Gil et al. created a thermoresponsive gel by combining gelatin and silk fibroin [36]. Silk fibroin crystals stabilized the gel at 37 °C. The gel swelled more at physiological temperatures than at 20 °C, but it also lost more mass due to gelatin disintegration and was released [48].

#### 2.2.3. Synthetic polymers and their derivatives

##### *N*-Isopropylacrylamide(pNiPAAm)-Based Systems

Thermoresponsive hydrogels based on pNiPAAm and its derivatives have been investigated extensively for drug delivery, cell encapsulation and cell culture surfaces. Coughlan and colleagues observed that with Hydrogel, the swelling was decreased in the presence of hydrophobic drugs, and the opposite effect was observed for hydrophilic drugs. The authors suggested that drug properties such as solubility, size, and chemical nature should be considered when choosing pNiPAAm hydrogel as a delivery vehicle [49]. The thermoreversible p(NiPAAm-co-AA) hydrogel was tested as a cell and drug delivery vehicle [50].

Liu et al. have polymerized p(NiPAAm-co-AA) with ethyl acrylate (EA). The swelling was lower due to the hydrophobic pEA. P(NiPAAm-coAA) showed an initial burst release, not observed on p(pNiPAAm-co-AA)/pEA. It was concluded that the pEA chains had a favorable effect on maintaining a slower and more stable release profile [51]. Yin et al. synthesized copolymers using NiPAAm and PAA. They showed that even small changes in pH could significantly affect the hydrogel thermo-responsiveness. This feature can be useful for applications such as drug delivery, where physiological temperature and local pH differences can both act as stimuli, and for molecular switching over the desired pH range [52]. Xu et al. formed a triblock copolymer hydrogel that showed a combination of stimuli-responsive attributes. A poly((2-dimethyl amino) ethyl methacrylate-co-2-hydroxyethyl methacrylate)-b-poly(*N*-isopropyl acrylamide)- b-poly((2-dimethyl amino) ethyl methacrylate-co-2-hydroxyethyl methacrylate) or p(DMAEMA-co-HEMA)-b-p(NiPAAm)-b-p(DMAEMA-co-HEMA) copolymer was synthesized by atom transfer radical polymerizations (ATPR). Its temperature-responsive behavior was attributed to pNiPAAm and the pH sensitivity to pDMAEMA [53]. A copolymer of pentaerythritol monostearate diacrylate (PEDAS), *N*-isopropyl acrylamide (NiPAAm), acrylamide (AAm), and 2-hydroxyethyl acrylate (HEA) was synthesized. PEDAS contains a lipophilic side chain, and AAm and HEA can modulate hydrophilicity and add groups for subsequent acrylation and crosslinking. The thermal gelation is attributed to the NiPAAm block. Future work is directed to be addressed biodegradability. Many homo-and copolymers of NiPAAm are not biodegradable, which may prove problematic for some biomedical engineering applications [54]. Nakayama and colleagues have prepared thermally responsive, biodegradable micelles that incorporate water-insoluble drugs. By combining a poly(*N*-isopropyl acrylamide-co-*N*, *N*-dimethyl acrylamide) (p(NiPAAm-co-DMAAm)) block, which has an LCST around 40 °C, with poly(D, L-lactide), poly(ε-caprolactone) or poly(D, L-lactide-co-ε-caprolactone), which are all biodegradable and hydrophobic, the group was able to fabricate polymeric micelles with controlled dimensions and phase transition temperatures [55]. Recently, Hatakeyama et al. have produced bioactive, thermoresponsive cell culture surfaces by immobilizing the cell adhesive peptide RGDS and the growth factor insulin on a NiPAAm-copolymer. *N*-Isopropylacrylamide was copolymerized with its analog 2-carboxy isopropyl acrylamide, and the polymer was grafted onto polystyrene tissue culture dishes, followed by RGDS and insulin immobilization. They found that these factors increase cell adhesion and proliferation, reducing culture time. When the temperature was brought to 20 °C, the cells could be quickly recovered as contiguous tissue monolayers [56].

#### 2.2.4. PEO/PPO-Based Systems

Triblock copolymers poly (ethylene oxide)-b-poly (propylene oxide)-b-poly (ethylene oxide) (PEO-PPO-PEO), also known as Pluronics^®^ or Poloxamers, exhibit a thermoreversible behavior at physiological temperature and pH [50]. The amphiphilic polymer structure has hydrophilic ethylene oxide and hydrophobic propylene oxide. The gelation occurs as changes in micellar properties function of both concentration and temperature. Above critical micelle concentration (CMC), the amphiphilic block copolymer molecules can self-assemble into micelles. Pluronics^®^ polymer drug delivery exhibits a CMC of 1 μM to 1 mM at 37 °C. Polypropylene oxide is relatively soluble in the water below critical micelle temperature (CMT). With a temperature increase, polypropylene oxide chains become less soluble, resulting in micelle formation [57]. The commonly used Pluronics^®^ in biomedical applications is F127 has a weight percentage of 70% PEO and a molecular weight of PPO around 4000 [58]. Pluronics have been extensively used in drug and gene delivery, inhibition of tissue adhesion, and burn wound covering. Pluronics^®^ can be a suitable substrate for hematopoietic stem cells, supporting their culture and preservation and tissue engineering applications. F127 was also evaluated as a scaffold for lung tissue engineering, showing promising results on tissue growth with a low inflammatory response [59]. Wein and colleagues tested a β-tricalcium phosphate (β-TCP) scaffold, using an F127 hydrogel to facilitate cell delivery and distribution for an in vitro study aiming at bone regeneration. They reported that F127 was no longer present in the channels of the β-TCP scaffold after one week in culture and seemed to have degraded. Bone tissue growth was only weakly induced, and the constructs showed lower stiffness than other hydrogels (fibrin, collagen I) composites evaluated [60]. Cohn and colleagues proposed two new mechanisms to create copolymers based on Pluronic^®^ F127 with improved mechanical properties. In both cases, they relied on the principle of a multiblock backbone with the addition of covalently bound repeating units. Both newly synthesized polymers exhibited significantly higher viscosities than F127 at 37 °C, and the poly(ether-urethanes) displayed much slower drug release kinetics than the original polymer [61]. Pluronic^®^ polymers were functionalized with acrylic moieties and thiols at their end groups and were subsequently gelled at 37 °C by Cellesi et al. It was found that these polymers were biocompatible, allowing for the encapsulation of sensitive drugs and cells. A similar method was followed with Tetronic^®^ polymers, which are thermosensitive, tetra-armed Pluronic^®^ analogs. By adjusting the molecular weight of the precursors and the functionalization (therefore also the crosslinking density), the final mechanical and transport properties of the “tandem” polymers can be controlled. Moreover, the “tandem” method allows for easy processing of the polymers, e.g., into spherical beads and hollow capsules [62].

#### 2.2.5. PEG/Biodegradable Polyester Copolymers

Thermoresponsive properties were ascertained by adjusting the hydrophobic polyester block and the PEG block length appropriately. These polymers were biocompatible, biodegradable, and exhibited a sol-gel transition. The use of high molecular weight-PLGA combined with low molecular weight-PEG resulted in a hydrogel with quick gelation at physiological temperature. The combination of hydrophobic/hydrophilic units created a surfactant behavior of the polymers in water, thus facilitating the solubilization of hydrophobic drugs. In vivo studies showed good mechanical properties and integrity for longer than a month [63]. More recently, Chen et al. developed a triblock PLGA-PEG-PLGA-based system for the controlled release of testosterone. Testosterone is water-insoluble, and so far, its delivery systems have included patches, creams, gels, injectables, and implants. A slower in vitro release of testosterone was observed for copolymers with longer PLGA blocks, possibly due to the slower degradation of these hydrophobic units. The thermosensitive polymers showed a controlled, linear release for three months [64]. Another recent approach towards a thermoresponsive system involved the synthesis of a multiblock copolymer with a biodegradable polyester. Alternating multiblock poly (ethylene glycol)/poly(L-lactic acid) (PEG/PLLA) copolymers were produced. It was shown that sol-to-gel transition depends on the total molecular weight (MW) and the MW of each building block. In vitro and in vivo gelation studies determined that a copolymer with a total MW of 6700 daltons and 600/1300 (MW of PEG/PLLA blocks, respectively) holds potential as an injectable carrier for biomedical applications in terms of transition temperature and modulus at 37 °C [65].

#### 2.2.6. Poly(organophosphazenes)

Current advances in poly(organophosphazenes) include their use as a drug [66] and cell delivery systems. Poly(organophosphazenes) grafted with mPEG and amino acid esters were reported as a new class of biodegradable and thermosensitive polymers in 1999 [54]. Sohn and colleagues developed a correlation for the LCST of these polymers as a function of their molecular structure, which comprises hydrophilic (PEG) and hydrophobic (amino acid esters) side groups. Polymers showed a sustained release profile for both hydrophobic and hydrophilic drugs. Moreover, their use as cell carriers holds promise. More recently, hydrogels exhibiting a thermosensitive sol-gel behavior have been reported as cell carriers for tissue regeneration. Typically, aqueous solutions of hydrogels used in biomedical applications are liquid at ambient temperature and gel at 37 °C [37,67].

## 3. Method of Functional Thermoresponsive Hydrogel Synthesis

Hydrogels can be made through physical or chemical processes depending on the application trigger. The interplay of oppositely charged particles or oppositely charged multivalent ion/surfactants with polyelectrolytes can produce physical hydrogels [68]. On the other hand, chemical hydrogels are usually made up of a covalently bonded polymer backbone. Such intelligent hydrogels may expand and shrink in reaction to variations in exterior environmental stimuli in a reversible manner. These hydrogels are composed of homo-polymeric, co-polymeric, or multi-polymeric networks, which comprise one, two, or more polymers. As a result, such hydrogels can perform multiple functions [3,69]. These multifunctional hydrogels can be made in any required dimensions. Adjusting their chemical composition, bioactivities, degradability, and numerous physicochemical features such as mechanical and rheological, spectral, pH stability, release, and loading characteristics, for instance, can be used in biological applications [70]. External factors that can form smart hydrogels are classified in Figure 1 [3].

### 3.1. Bulk Polymerization

Because of its easy method, bulk polymerization is commonly used to make hydrogels. This procedure employs a modest quantity of crosslinkers to polymerize liquid monomers and monomer-soluble initiators. UV light, irradiation, and/or chemical accelerators are commonly used to start the polymerization [3,6,68]. Bulk polymerization does have a faster polymerization speed and ineffective temperature control, and the solution viscosity rises quickly. Controlling this conversion is thus critical for tuning the hydrogel characteristics. The conversion of bulk polymerization can be controlled by adjusting the temperature and initiator amount. Another solution is to stop this reaction at a low conversion level, although this is regarded as unprofitable in large-scale systems [68]. As a result, various polymerization approaches for hydrogel fabrication, such as solution, suspension, and emulsion polymerization, are frequently used.

### 3.2. Solution Polymerization

Ionic or neutral monomers with a solvent (e.g., benzyl alcohol, water, ethanol, or water-ethanol mixes) and versatile cross-linkers are polymerized by ultraviolet or redox activation to create hydrogels in solution polymerization. The hydrogel is isolated, and any remaining monomers, cross-linkers, initiators, or other contaminants are washed away using distilled water [3,71]. The advantages of solution polymerization include simplicity of synthesis, relatively low costs, and higher heat transfer control throughout the polymerization. Polymerization is safe and harmless because it takes place in an aqueous phase. For the production of cellulose-based superabsorbent hydrogels, solution polymerization is widely utilized. The polymerization rate is fast, and the process can be carried out at ambient temperature. Because the solution has low viscosity, agitating the reaction mixture is easy. As a result, solution polymerization achieves better heat transfer and dissipation than bulk polymerization [36,71,72].

### 3.3. Suspension Polymerization (Including Inverse-Suspension Polymerization)

Suspension polymerization uses insoluble monomers and initiators into an aqueous phase with a minimal hydrophilic–lipophilic equilibrium suspending agent. These are continuously stirred to produce monomer drops ranging from 0.1 to 5 mm in diameter [3,73]. These polymeric hydrogel beads develop during polymerization, and they can be sifted to remove from the reaction medium. Individual monomers are bulk polymerized on a small scale. Since water is the most common medium, it is an excellent heat transfer medium. To prevent the droplet from coalescing, a colloidal protecting agent, e.g., carboxymethyl cellulose (CMC) or methylcellulose (MC) and polyvinyl alcohol (PVA), is frequently utilized. Inverse suspension polymerization is also commonly utilized in the production of hydrogels [74].

### 3.4. Emulsion Polymerization

Emulsion polymerization can be used to make hydrogels. A water-soluble initiator, a surfactant, crosslinkers, and actual but tiny water-soluble monomers are used in a conventional emulsion polymerization process (i.e., slightly water-soluble or completely hydrophobic monomers). A hydrophilic monomer from an organic solvent is employed in inverse emulsion polymerization [65]. It creates polymer molecules that are substantially smaller (0.1–3 m) than those produced by suspension polymerization. Heat flux can be accomplished successfully over bulk polymerization, and the suspension and emulsion polymerization process can be easily manipulated. Emulsion polymerization, unlike suspension polymerization, utilizes a water-insoluble initiator [68,75].

### 3.5. Hydrogel Synthesis by Chemical Mechanism

#### 3.5.1. Chain Growth Polymerization

Chemically crosslinked hydrogels are commonly made using a free radical process of chain-growth polymerization. Initiation, propagation, and termination are the three phases of polymerization. In radical polymerization, most monomers include hydrophilic unsaturated alkenes, alkynes, and aromatic compounds containing alkenes and styrene based monomeric groups [69]. Recently, Lee et al. demonstrated that chain growth polymerization-based PEG hydrogel showed improved protein release efficiency and diffusivity with decreasing PEG concentration. Interestingly, the molecular weight of PEG had no significant impact on protein release efficiency and diffusivity [76].

#### 3.5.2. Graft Polymerization Mechanism

By adopting a grafting process, poor mechanical characteristics of a bulk polymerized hydrogel were obviated, particularly when grafting onto more strong support frames [68,69]. Free radical regions are typically formed on the support surfaces, where monomers could be immediately polymerized to produce strong covalent connections with the framework. Grafting vinyl monomers onto polysaccharides is a frequent practice. Furthermore, grafting PAAc from hybridized chitosan (CHT) with cellulose using thiourea formaldehyde glue results in pH-responsive hydrogels that are structurally more robust than grafted AAc from CHT hydrogels [66].

#### 3.5.3. Step-Growth Polymerization

To make hydrogels, step-growth polymerization uses various functional groups comprising monomers. To promote a one-step polymerization process, the complimentary functional groups interact and create covalent bonds [68]. The mechanical characteristics of photodegradable hydrogels made by chain and step-growth polymerization techniques were investigated by Tibbitt et al. Because of network uniformity and cooperativity, step-growth hydrogels have greater mechanical stability, tensile strength, elasticity, and shear strain to yield than chain-growth hydrogels [77]. On the other hand, chain-growth hydrogels have a lower rate of light-induced degradation due to their increased network connection. Desired hydrogel characteristics can be attained by comprehending hydrogel networking [78].

#### 3.5.4. Crosslinking Method

Physically crosslinked hydrogels generate a network in polymeric materials via physical processes such as ionic interactions between polycations/multivalent cations and polyanions or hydrophobic interactions among polymer chains [3]. On the other hand, chemically crosslinked hydrogels generate polymeric by chemical bonding (i.e., covalent bonds) [79]. Crosslinking can be induced here by heating or ultraviolet light, and chemical crosslinking is produced through various reactions, including the Michaelis–Arbuzov reaction, nucleophilic reaction, and Michael’s reaction etc. Different applications of hydrogels are feasible by optimizing their characteristics, which are governed by crosslinking degrees [80].

#### 3.5.5. Synthesis of Thermo-Responsive Hydrogels (TRHs)

Temperature-responsive hydrogels are among the most intelligent types of hydrogels because they can modify their structure and formation in reaction to temperature fluctuations. The inclusion of hydrophobic groups, such as methyl, ethyl, and propyl groups, distinguishes them. TRH gels are often classified as positively thermosensitive, negatively thermosensitive, or thermally reversible [81]. Positively thermo-sensitive hydrogels, such as interpenetrating polymeric networks (IPNs) based on poly (acrylamide-co-butyl methacrylate) [P(NIPAAm-co-BMA)] and PAAc, expand at extreme temperatures and contract at low temperatures [3,6,82]. On the other hand, negative temperature-dependent hydrogels swell when the temperature decreases and vice versa. Biopolymers (such as chitosan, cellulose, and gelatin) and synthetic materials (such as poly(*N*-isopropyl acrylamide) (PNIPAAm) and polyfluorene 127) can be used to make TTRHs. The kind of monomer and crosslinker used in producing a hydrogel is regulated primarily by the kind of monomer and crosslinker employed [1,4,6].

## 4. Mechanism of Thermoresponsive Hydrogel

Thermoresponsive hydrogels originate from phase transition from gel phase to solution-phase and vice versa with temperature variation. The interaction of surrounding phase and functional group of polymers important for gel formation. In the case of a thermoresponsive hydrogel, the cross-linking ability has achieved widespread biomedical application. Poly(*N*-isopropyl acrylamide) (PNIPAM) is an ideal natural thermoresponsive polymer [83]. The temperature at which a functional copolymer undergoes the transition from solution phase to gel phase is called a sol-gel phase transition. One class of functional polymeric material transformed to a solid phase (gel-like consistency) and separated from the solution above a specific temperature threshold limit, defined as the lower critical solution temperature (LCST). However, another class of functional polymer exhibited a solid phase (gel-like consistency) after cooling their solutions. Another threshold limit is upper critical solution temperature (UCST) [67,84].

Thermoresponsive polymers which solubilize in an aqueous solvent or organic solvents are not suitable for biomedical applications. As shown in Figure 2. LCST polymers are soluble below a critical temperature, whereas UCST polymers solubilize above UCST. The water in thermo-responsive hydrogels is altered by small changes in temperature and the resulting polymeric chain transformation from hydrophilic to hydrophobic above low critical solution temperature (LCST) is called hydrophobic hydration [85].

### 4.1. LCST Polymers

In the category of LCST polymers, poly-*N*-isopropyl acrylamide (PNIPAAm) is a widely studied polymer. The literature reports pioneering work on physical-chemistry aspects of solution properties of PNIPAAm and LCST polymers. These are vinyl polymers synthesized by radical polymerization with secondary amide pendant groups. The lone pairs present on the oxygen atom and the lone pair of the nitrogen atom of the amide bond serve as hydrogen bond acceptors. In contrast, the hydrogen covalently attached to the nitrogen atom is a donor of hydrogen bonds. These results inter and intramolecular hydrogen bonding with water (Figure 2). LCST is a reversible phenomenon when the temperature of the polymeric system decreases, the hydrophilic-hydrophobic balance is regained for higher hydrophilicity, and the polymers become soluble [86,87]. Moreover, due to structural changes of the polymer network, polymeric systems with LCST show a coil-globule (C-G) transition. That implies that the free Gibbs energy (∆*G* = ∆*H* − *T*∆*S*) of dissolving polymer in water is negative at reduced temperatures and positive at elevated temperature from a thermodynamics standpoint. If the enthalpy of hydrogen bonding among water molecules and polymer chains (∆*H*) and the entropy contribution (∆*S*) are both negative, such behaviour is feasible. That means, if water loses entropy as it hydrates the polymer chains, the entropy term (−*T*∆*S*) will began to dominate the operation as the temperature of the solution rises, leads to a positive Gibbs free energy of mixing [88]. Phase separation will occur as a result of this. This shift is usually reversible, allowing for a rapid, reversible, thermally induced phase change. Poly (*N*-isopropyacrylamide) was the first to be discovered and has been the subject of the most research [85,87].

PNIPAAm copolymers showed cloud points around 32 °C, with polymer concentrations ranging from 5–30 wt.%. Polymer with LCST 32 °C easy to design formulations with properties from room temperature to 37 °C. In the scientific community, there is growing interest in incorporating polymeric blocks in PNIPAAm’s to turn PNIPAAm’s LCST towards higher temperature for exploring hyperthermia applications for targeted disease areas and drug release observed by the disruption of polymeric architecture. A polymer chemist explored this area by incorporating hydrophilic monomer moiety to shift the hydrophilic/lipophilic balance towards more hydrophilic. This results in a greater interaction with aqueous solution as well as a higher temperature required to achieve LCST for disorganization of polymer and release of internal encapsulated cargo [89]. In the literature, the PNIPAAm block copolymer-based micelle was synthesized with different hydrophobic blocks of a polymer, such as a polystyrene and poly-L-lactide (PLA), to encapsulate doxorubicin. These LCST polymeric micelles destabilize at 37–42.5 °C and release doxorubicin, unlike at 37 °C for PNIPAAm [90].

### 4.2. UCST Polymers

Once the solute-solute and solvent-solvent interactions overwhelm the solute-solvent interaction to yield a positive enthalpy of mixing, the UCST phase change occurs. Assume the enthalpic term (∆*H*) in the Gibbs free energy expression to indicate the supramolecular association of the polymer. However, the complex molecular interaction strength diminishes with rising temperature, causing the hydration component to prevail and contribute to polymer breakdown. To achieve the UCST shift greater than the theoretical LCST shift, which can lead to total insolubility, the polymer has to be hydrophilic in nature. The reaction between poly (acrylic acid) with poly(acrylamide) in water, driven by hydrogen bonding between the carboxy and amide groups, is an example of a UCST-type polymer [88,91].

These polymers are soluble above UCST and insoluble below the critical temperature. Unlike LCST polymers, UCST polymers are less explored for aqueous systems due to some limitations, but nowadays, there is growing interest amongst the scientific community for UCST polymers. Thermoresponsive in the UCST-based copolymers depends on the hydrogen bonds or electrostatic interactions between functional groups with the surrounding aqueous environment. In this class, polymers that exhibited zwitterion and ion exhibited this phenomenon [8]. Strong supramolecular interactions (hydrogen bond, electrostatic bond in zwitterion ions) between groups of copolymers with their surrounding aqueous conditions resulted in UCST, as shown in Figure 3. poly(*N*-acryloyl glycinamide) polymers are non-ionic polymers shown UCST behavior in water by virtue of hydrogen bonding between the polymer side groups [92,93]. Figure 4 shows thermoresponsive behaviors in the UCST copolymers achieved by hydrogen bonds between poly(*N*-acryloyl glycinamide) polymer chains and electrostatic bonds between zwitterionic groups in poly(*N*, *N*′-dimethyl(methacryloylethyl)ammonium propane-sulfonate).

### 4.3. Mechanism of Phase Transition in Thermoresponsive Hydrogels

Multiple mechanisms reported in the literature for thermoresponsive UCST and LCST copolymers resulted from phase transition in an aqueous environment. The primary mechanism is the hydration of UCST/LCST copolymers by intra and intermolecular hydrogen bonding, which affects the solubility of copolymers resulting in a change in information of hydrogels with respect to change in temperature. Factors responsible for phase transition in the thermoresponsive hydrogels are ionic interactions (electrolytes), Van-der-Waals interactions, hydrophobic interactions and hydrogen bonding. Other factors contributing to phase transition are the interaction of the functional group of copolymer with the surrounding environment, thermodynamics, and the negative energy of the system [95,96].

## 5. Mathematical Models for Drug Release from the Hydrogel-Based Formulations

Many hydrogels and stimuli-responsive hydrogel-based drug delivery have been developed and fabricated to fulfill the dire need of the pharmaceutical development and medical field. Mathematical modeling plays a crucial role in developing hydrogel networks, by which one can decide and identify critical parameters and drug release mechanisms. Moreover, this could provide some basic and essential understanding in the development of control release hydrogel formulations and their release kinetics estimation. The drug release is a critical parameter in developing all kinds of matrix formulations. In the case of hydrogels, the physiochemical properties of the hydrogel and drug loading method in the hydrogel network could decide the drug release mechanism from the cross-linked matrix [97]. The drug loading into the hydrogel delivery systems matrices could be done through the two following methods:**Post-loading (drug)**

The drug loading will be performed after the hydrogel network is formed. The drug release mechanism is based on the gradient systems or driving force. In the case of an inert hydrogel system, the primary driving force for drug uptake is the diffusion process, and drug release will be determined by the diffusion and gel swelling index. In the case of hydrogel containing drug-binding ligands, drug–polymer interaction and drug diffusion could be considered and included in any drug release models [98].


**In-situ drug loading**


In this process, drug or drug-polymer conjugates are thoroughly mixed with polymer precursor solution, and the hydrogel network development and drug encapsulation/loading are done simultaneously. This is a matrix hydrogel system, and the drug release is controlled by a diffusion process, hydrogel swelling, reversible drug-polymer interactions or labile covalent bond degradation [97,98].

Broadly the drug release from hydrogel is classified into three mechanisms as follows:Diffusion-controlled hydrogel drug delivery systemsSwelling-controlled hydrogel drug delivery systemsChemically controlled hydrogel drug delivery systems

### 5.1. Diffusion-Controlled Hydrogel Drug Delivery Systems

It is essential to understand the drug release mechanism and crucial parameters that govern the drug release from the different types of hydrogel systems, which could help precisely determine the drug release profile. The drug release depends on the nature of the hydrogel, i.e., porous or non-porous. In the case of porous hydrogel systems, generally, the pore size is larger than the molecular drug size. Drug diffusion coefficients would depend upon the porosity and tortuosity of the hydrogel systems [99]. Although, in the case of a non-porous hydrogel system, the drug diffusion coefficient is relatively low because of the matrix nature of the gel system. The steric hindrance generated by the polymer chains within the polymer cross-linked or matrix systems could decrease the diffusion coefficients [3,4,5]. Generally, in these cases, there is a decrease in average free volume available per molecule, and the hydrodynamic drag experienced by the drug is increased, leading to an increased drug diffusion path than the porous hydrogel systems with pore sizes greater than the loaded or encapsulated drugs. Because of the high permeabilities of the hydrogel systems and the benefits of in-situ fabrication, most research has focused on the basic understanding of the diffusion-controlled release drug release mechanism from the three-dimensional hydrogel matrix systems [5,6].

Fick’s law of diffusion or Stefan–Maxwell equations explain the drug diffusion through highly swollen hydrogel systems. The diffusion-controlled hydrogel system could be a reservoir or matrix type system [100,101,102,103]. In the case of a reservoir system, the drug is deposited in the center, and it is by the polymeric layer of hydrogel membrane, so the drug release kinetics could be explained based on the Fick’s first law of diffusion because that deals with membrane release.
(1)$$JA=-D\frac{dCA}{dx}$$
where *JA*: drug flux, *D*: drug diffusion coefficient, and *CA*: drug concentration

In so many cases, the molecule diffusion coefficient is considered constant, which helps to simplify the modeling. Moreover, it is generally a function of the drug concentration, so concentration-dependent drug diffusivity must be used to determine drug flux accurately. Another assumption of this expression is that *JA* is the drug flux related to the average mass velocity of the system.

In the case of the matrix system, the drug is uniformly distributed through the polymeric hydrogel systems. The controlled and unsteady state of drug diffusion in a one-dimensional matrix (slap-shaped) could be explained using Fick’s second law of diffusion;
(2)$$\frac{dCA}{dt}=D\frac{d^{2}CA}{dt^{2}}$$

Again, in this case, the drug diffusion coefficient is considered constant. Other assumptions include sink conditions and a thin planar geometry where the release through slab edges is neglected. If the drug diffusivity is concentration-dependent, the following equation is used:(3)$$\frac{dCA}{dt}=\frac{d}{dx}\;\left(D\;\left(CA\right)\frac{dCA}{dx}\right)$$

Scientists have attempted to predict and model the diffusion control drug delivery from the hydrogel systems, mostly empirically determined drug diffusion coefficients. Once the *D* is determined using Equations (1) and (2). This could be solved based on proper initial boundary conditions to provide a drug concentration profile that suggests release kinetics. For example, separating variable techniques could obtain an exact analytical solution to Equation (3). The ratio of the amount of drug released up to any time *t* (*Mt*) to the final amount of drug release (*M*∞) could be expressed as:(4)$$\frac{Mt}{M\infty }=1-\sum_{n=0}^{\infty }\frac{8}{{\left(2n+1\right)}^{2}\pi^{2}}\;\exp\left[\frac{-{\left(2n+1\right)}^{2}\pi^{2}D}{L^{2}}t\right]$$

This equation could determine the diffusion of the small (synthetic) and larger molecules (biomolecule, i.e., protein, peptide, and monoclonal antibodies) based on the obtained diffusion coefficients. However, this simple equation or solution is used for most diffusion-controlled drug release systems. Complex modeling would increase the other mechanisms. It involves different types of interactions, such as polymer-drug interactions, which mostly happen in a non-spherical drug used in the hydrogel systems.

Another empirical equation proposed by Peppas et al. considers a time-dependent power-law function in the drug release mechanism [8,9].
(5)$$\frac{Mt}{M\infty }=k.t^{n}$$
where *k* denotes the structural/geometric constant for a specific system and n the release exponent, indicating the release mechanism.

Table 1 gives the n values for the drug delivery matrix and several geometries and drug release mechanisms [97]. The fractional drug release (*Mt*/*M*∞) appears to be zero-order in a purely swelling-controlled slab-based delivery system as the release exponent equals unity. The power law is quite simple for most diffusion-controlled drug delivery systems from the matrix systems. It offers a very accurate and robust drug release prediction from complicated systems. In the case when n = 0.5 in a diffusion control system, this power-law is valid for the first 60% drug release profile. This model only predicts drug release profiles after performing the release experiments. The experimental data are used to determine the drug release mechanism. However, it could have certain limitations. It won’t give details after specific chemical or network system changes in the formulation. That means it does not provide an idea about the drug release mechanism when the chemical, network, or matrix system type changes.

In the case of the hydrogel drug delivery systems, drug diffusivity purely depends upon the polymer swelling index and cross-linking density of the hydrogel. Hence, the diffusion coefficient could be used to explain whether the drug release would be sensitive to outer or inner environment changes or polymer degradation over time in the drug delivery system. In the literature, several theoretical models have been proposed that explain the drug release and its coefficients to fundamentally understand the behavior and characteristics of the hydrogel systems [84,100].

Generally, theoretical models for predicting drug diffusion coefficients have the following general form:(6)$$\frac{D_{g}}{D_{o}}=f\left(r_{s},V_{2S}§\right)$$
where
*D_g_* is drug diffusion coefficients in the swollen hydrogel network or matrix*D_o_*, the drug diffusion coefficients in pure solvent*r_s_*, the size of the drug to be delivered

This general equation considers factors affecting the drug release from the hydrogel systems as follows:Gel structurePolymer compositionSize of the molecule/drugWater content in the systems

For a degradable hydrogel system, *D_g_* changes as the polymeric matrix or network degrade because of the increase in the gel mesh size and a decrease in polymer volume fraction during the release. Different theories have explained the relationship between the drug diffusion or diffusivity in the hydrogel and the solution systems [100]. Lustig and Peppas have proposed the equation based on the free volume approach that has explained the relationship between drug diffusivity and hydrogel network structure [105];
(7)$$\frac{D_{g}}{D_{o}}=\left(1-\frac{r_{s}}{§}\right)exp\left(-Y\left(\frac{v_{2s}}{1-v_{2s}}\right)\right)$$
where *Y* is the ratio of the critical volume needed for a translational movement of the encapsulated molecule and the average free volume per solvent molecule. Generally, a good approximation for *Y* is considered as unity. For the highly swollen hydrogel system Q > 10, with degradable polymer cross-links, the diffusivity correlation is shown in Equation (7) and can further be simplified during the early stages of the degradation of the hydrogel systems [106,107],
(8)$$-1\frac{D_{g}}{D_{o}}=\frac{r_{s}}{\text{§}}\;\sim e^{-7/5jk_{E^{\prime }t}}$$
where parameter *jk_E_*_′_ is the pseudo first order of the reaction rate constant for the hydrolysis of a prone cross-link group. This equation shows that the mesh size is time dependent because of polymer network degradation. *D_g_* increases as degradation proceeds and approaches *D_o_*. The increase in drug diffusivity depends on polymer network structure and cleavage kinetics of bonds [108].

### 5.2. Swelling-Controlled Hydrogel Drug Delivery Systems

Swelling controlled drug delivery is another type of system that undergoes a phase transition from a glassy state to a rubbery state. In the case of the glassy state, the drug molecule remains in the compact and immobilized state [106,107,108]. However, drug molecule diffusion increases in the rubbery state. The drug molecule release rate is depends on gel swelling in this system [106,107,108]. This swelling controlled hydrogel drug delivery is explained by the example of hydroxypropyl methylcellulose (HPMC). The drug-loaded HPMC tablet contains hydrophilic matrices, is three-dimensional, and the drug is present in the glassy state. HPMC polymer absorbs liquid after oral administration. A rapid glassy-to-rubbery phase transition occurs once the glass transitions temperature is attained, releasing the drug from the swellable hydrophilic matrix. The drug release rate is controlled by the gel layer thickness and rate of water transport. The drug delivery was determined by two critical parameters: drug diffusion time and polymer chain relaxation time. However, in the case of diffusion-controlled drug delivery, the swollen phase’s time-dependent thickness (δ*t*) is a rate-limiting step. In the case of swelling-controlled drug delivery systems, polymer relaxation time (ʎ) is the rate-limiting step. These two systems are compared by the Deborah number (*De*) on two-time scales [109,110].
(9)$$De=\frac{ʎ}{t}\;=\frac{ʎD}{\;\delta {\left(t\right)}^{2}}$$

In the case of a diffusion-controlled drug delivery system (*De* ≪ 1), the drug molecule dominates the fickian diffusion. This is explained in the previous section’s diffusion release mechanism, which could be used in drug release prediction. However, in the case of swelling controlled drug delivery system (*De* ≫ 1), the drug release depends upon the polymer network swelling rate. The empirical power-law Equation (4) could describe the diffusion-controlled drug delivery from the hydrogel system and be precisely used for swelling-controlled hydrogel drug delivery systems. This equation is further modified by considering drug diffusion and polymer relaxation [111].
(10)$$\frac{M_{t}}{M_{\infty }}=k_{1}t^{m}+k_{2}t^{2m}$$
where *k*_1_, *k*_2_, and m are constants

The two terms on the right side indicate the release profiles governed by diffusion and polymer relaxation contribution.

The above mathematical relationship could not consider the moving boundary conditions in that the gel expands heterogeneously once the water penetrates, leading to swelling of gels. Korsmeyer and Peppas derived the dimensionless swelling interface number, *Sw*, for a more rigorous understanding of moving boundary phenomena in hydrogel swelling [112,113,114].
(11)$$Sw=\frac{V\delta \left(t\right)}{D}$$
where *V* denotes the velocity of the hydrogel swelling front and *D* the drug diffusion coefficient in swollen phase.

For a slab system when *Sw* ≪ 1, drug diffusion is quicker than the movement of the glassy-rubbery interface, and therefore, a zero-order release profile is anticipated.

The more robust method of predicting drug release from the swelling controlled drug delivery system is explained by the sequential layer model proposed and developed by Siepmann and Peppas [105,115]. Three critical parameters were considered for this model: drug diffusion, dissolution, and polymer relaxation. Moreover, drug transports (axial and radial) were considered per the Ficks second law of diffusion for cylindrical geometry with the concentration-dependent diffusion coefficients shown below [112,116]
(12)$$\frac{dC_{K}}{dt}=\frac{d}{dr}\left(D_{K}\frac{dC_{k}}{dr}\right)+\frac{D_{k}}{r}\frac{dC_{k}}{dr}+\frac{d}{d_{z}}\left(D_{k}\frac{dC_{k}}{d_{z}}\right)$$
where *C_k_* and *D_k_* are the concentration and diffusivity of (1: water; 2: drug), respectively [117].

Concentration-dependent diffusivities derived by a “Fujita-like” free-volume model could be represented as follows,
(13)$$D_{1=D_{1eq}\;\dot{exp}}(-\beta_{1}\left(1-\frac{C_{1}}{C_{1eq}}\right)$$
(14)$$D_{2=D_{2eq}\;\dot{exp}}(-\beta_{2}\left(1-\frac{C_{1}}{C_{1eq}}\right)$$
where $\beta_{1}$ and $\beta_{2}$ are dimensionless constants, and “*eq*” is the equilibrium drug concentration at the water/matrix interface, where polymer disentanglement happens.

Because of the concentration-dependent diffusion coefficients, Equations (13) and (14) could be solved mathematically. Siepmann et al. showed that these mathematical solutions were established well with experimental results. This model is thus helpful in predicting the shape and dimensions of HPMC tablets required to obtain anticipated release profiles [110,118,119,120].

### 5.3. Chemically Controlled Hydrogel Drug Delivery Systems

Already we have discussed the diffusion and swelling controlled hydrogel drug release mechanism in the previous sections. The third type of drug release mechanism is chemically controlled hydrogel drug delivery. This is a crucial drug release mechanism further classified as follows: Pure kinetic controlled hydrogel drug release mechanism, where polymer degradation (bond cleavage) takes place. This is the rate-limiting step. However, diffusion is considered a negligible parameter in the modeling.Reaction diffusion-controlled hydrogel drug release mechanism. Both reaction (drug-polymer and protein-drug interactions and polymer degradation) and diffusion terms should be considered in the modeling to predicate the accurate drug release from the hydrogel systems. This mechanism is mainly considered in the interest of the synthetic hydrogel systems, which are developed and designed with drug binding capacity and are used in drug delivery, biomedical, and tissue engineering applications [121,122,123,124]. The kinetically and reaction controlled hydrogel drug release has been classified into different types, as follows in Table 2.

### 5.4. Miscellaneous Types of Hydrogel Systems and Release Mechanism

Several novel hydrogel drug delivery systems were developed for the advanced drug delivery of small biotherapeutics and biomedical and tissue engineering applications. These hydrogel systems release drug molecules through different drug release mechanisms. These hydrogel systems are complex systems that follow the multiple release mechanism-based chemical degradation, diffusion, and stimuli-based release. The different miscellaneous type of hydrogel systems and their drug release mechanism is given the Table 3.

### 5.5. Advanced Hydrogel Systems and Their Drug Release Challenges

Mathematical simulations have been performed for the drug release prediction from the complex hydrogel systems, an excellent development approach. However, many challenges are associated with predicting the accurate drug release profile from the hydrogel systems. The fundamental understanding of drug release is essential for setting appropriate mathematical models. In the drug release mechanism of the hydrogel systems, drug molecules are translocated from the inner to the outer environment of the systems. In this process, multiple critical factors and processes are involved, which could hamper the drug release, such as polymer cross-linking density, polymer swelling, gel degradation, size and charge on loaded or/encapsulated drug molecules and the drug-polymer interaction and other types of physical or chemical interactions. Moreover, in the specialized hydrogel system with a ligand bound with the polymer matrix or drug, the binding of the ligand would be considered, and that is quantified for accurate prediction of the drug release from the matrix hydrogel systems. Further, several other challenges in the accurate drug release prediction vary based on the hydrogel type and its complex nature [1,9,17,25].

## 6. Applications of Functional Thermo-Responsive Hydrogels

With recent advancements in biomedical sciences, hydrogels, as a novel class of formulations composed of functional polymers, have well proven their potential applications, e.g., nano-composite, injectable, topical, conductive, bio-sensing materials, drug delivery, etc. These advanced hydrogels exhibit excellent structural and mechanical properties and their applicability to overcome the limitations of conventional hydrogel formulations [139,140].

### 6.1. Applications in Biosensing

Hydrogels are now emerging as excellent tools in diagnostic and biomedical assays with their role in immobilization, embedding bio molecules, and responsive/functional material [141].

Chemical structure and polymer properties are the key factors determining the crucial parameters of hydrogel functionality, including sensitivity, response, target specificity, reproducibility, etc. [141]. The probes as biosensing elements are immobilized on a solid support, and it is chosen to interact specifically with the target analyte. Glucose monitoring devices and pregnancy test kits are typical examples of biosensors devices. These devices are applicable in healthcare sectors and are widely applicable in forensic sciences, environmental, food sciences, etc. [142]. The 3D hydrogel structures are more advantageous in biosensing applications as these can accommodate more number recognition elements and possess an aqueous and biocompatible microenvironment, which further leads to more stability and preserves functionality, specificity, and sensitivity [143].

Basic aspects of applicability of hydrogels as biosensors are their use as immobilization/encapsulation matrix, substrate coating, substrate replacement, and stimuli-responsiveness etc.

Compared to two-dimensional bio-interfaces, 3D structures give increased surface area and porous structures that allow immobilization of recognition elements at larger extents. Hydration gives structural similarity to biological tissues, which provides optimal substrate retention and biological interaction. The entrapment of probes inside the matrix will lead to comfortable accommodation without surface interaction which is responsible for maintaining biological response [143,144]. Immobilization also provides prevention of nonspecific binding due to antifouling polymers and solutions resembling the environment developed by hydrogels. The flexible nature of hydrogel provides enhanced accessibility of probes, high surface binding and good exposure to the recognition sites. Hydrophilicity provides significantly reduced nonspecific binding [145].

Hydrogels are also capable of replacing substrates other than the property of surface coating. This technique is multi-step, highly sensitive and needs amplification [146]. The intact structure of biomolecules needs to be preserved for proper sensitivity and specificity of hydrogels in bio sensing. Updike and Hicks made the initial attempt in 1967 [147] whereby the enzyme glucose-oxidase was entrapped in a gel of polyacrylamide, which provides stability and amplification in sensor preparation [147,148]. Various other approaches for encapsulation, such as size-selective encapsulation and covalent linkage, also have good potential to carry probes.

These will give excellent loading capacity and accessibility. Some attempts have been reported for enzymes to hydrogels to preserve intact protein structure, leading to enhanced enzyme activity [149,150]. In a flow-through electrochemical sensor, super-porous agarose gel is used to immobilize the signal-producing enzymes, which will increase the signal intensity with retention of good flow [151]. Stimulus responsive hydrogels based upon stimuli like pH change are mainly composed of a high extent of carboxylic group in polymers [152,153]. Upon increased pH, these carboxylic groups become charged and lead to swelling due to charge repulsion [154]. The extent of swelling can be detected. The example enzyme urease was encapsulated into a pH-responsive hydrogel, and in the presence of an amount of urea with increased pH, hydrogels swelled. Further, suitable pressure sensor device estimation is carried out [155]. Hydrogels provide an enhanced loading capacity, prevent nonspecific adsorption, and reduce background signals, modulations, high aqueous environments, as well as biocompatibility [141,156,157], etc.

### 6.2. Applications in Drug Delivery

Temperature responsive polymers have excellent potential in biomedicine, enabling cell layer production, in situ drug delivery, and 3D printing under physiological conditions. Research on temperature-responsive polymers has made great strides in recent years. There are various research opportunities, such as surface generation and 3D printing, and further translation applications are possible in the near future [158]. In tissue engineering and regenerative medicine, it supports cell adhesion and unique bio imitation. Naturally, derived materials have been significantly impacted because of their biocompatibility and mimic extracellular matrices. Novel approaches in hydrogel developments have emerged to meet the specific needs of tissues used in the latest technology and take advantage of the unique properties of nature-based materials. Tissue engineering hasemerged by providing functional tissue and organ replacements. Over the last few decades, significant scientific advances have been made in cell biology and biomaterial [158]. Moreover, when using hydrogel as a substrate for bone tissue engineering or medication delivery (once it gets into contact with biological fluids), it is vital to look into the swelling and deswelling characteristics of the hydrogel formulations. Shah et al. (2015) investigated the effect of simulated biofluids on the swelling ratio of a biomineralized (CaCO_3_) PVP–CMC hydrogel in the existence of additional different biological fluids: glucose solution, physiological solution, and urea solution, all at the same pH (7.5) and temperature (37 °C). The hydrogel in glucose solution demonstrated that the interaction of glucose substituent with the mineralized hydrogels increases slightly the charge density inside the hydrogels, thus increasing the hydrophilic nature, which eventually results in a greater swelling ability, however there is a constant uptake of physiological solution by the hydrogel, but after a certain time interval the absorption has become steady. Physiological solution must resist osmotic pressure within the gel throughout swelling tests, and when osmotic pressure falls, water penetrates into the gels quickly, causing the gel to expand Additional hydrophilic sites, such as NH_2_^+^ and C=O, are available in urea solution, and urea, as a weak base, can easily conjugate with the COOH group of cellulose in the hydrogel. As a result, urea develops increased hydrophilicity when it combines with water and causes the biomineralized (CaCO_3_) PVP–CMC hydrogel to swell. In this case, super saturation was reached in 150 min. Two major aspects are emphasized on when it comes to swelling behavior: (1) Donnan osmotic pressure and (2) elastic property inside the polymeric system structure. When these two events are equivalent, the material can no longer absorb the liquid and the supersaturation state is reached.

Nowadays, advancements in technologies concerning bio link improvement with cyto-compatibility are emerging in 3D printing techniques [159,160]. Thus, new polymeric biomaterials that may triumph over those obstacles are especially needed, making reversible hydrogel structures attractive options. Moreover, every other capacity approach to deal with the shortage of bio-inks is the aggregate of a thermo-responsive gelatin community, which gives remarkable extrusion and structural balance all through 3-d printing [161]. Applications of novel polymeric material networks with gelation properties help amplify bio links in tissue engineering and regenerative medicine [162]. Regenerative medicine has excellent potential for better treatment outcomes triggered by advanced bioengineering approaches over a few decades. Tissue engineering strategies can retain, restore, and revitalize lost tissue and organs. Various methodologies used to integrate bioactive constituents, biomimetic materials, and cells play a crucial role in promoting tissue regeneration. The hydrogel approach has proven its potential over the past two decades in tissue engineering, which has excellent potential to maintain 3D structure, providing mechanical strength and stimulation of extracellular matrix. Due to high water content, these hydrogels provide an excellent environment for cell survival and act as mimic structures for native tissues. This system also serves as a matrix for cell immobilization and delivery of growth factors [163]. For a long time, a paradigm shift has been under way in the design, fabrication strategies, homes, and packages of clever hydrogels in biomedical engineering. This may be attributed to massive studies performed in growing state-of-the-art primarily hydrogel-based matricing structures within the subject of regenerative medicinal drugs. Current studies are targeted mainly at altering biopolymer chemistry. Repeated efforts were made to tailor clever hydrogels for particular packages by cautiously investigating their surrounding microenvironments and using those purposeful enhancements for additional exploitation [164,165]. Finally, despite large improvements in hydrogel fabrication strategies, various parameters, such as the rate of degradation rates, time of response, floor hybridization, microstructures and the immunological reaction of those substances call for a cautious evaluation to synthesize greater cytocompatible hydrogels for tissue engineering packages. Recently published works employing thermoresponsive hydrogel for drug delivery were summarized in Table 4. In the future, additional developments in shape and characteristic features will be able to facilitate interaction studies and modular approaches in the generation of new tissues [166,167,168,169].

### 6.3. Applications in Self-Healing

These are types of hydrogels that possess the ability to recover and are used in various fields. These are composed of soft materials with recoverable properties. These also have a variety of advantages, including injectability, adhesiveness, conductivity, applications as drug/cell delivery vehicles, tissue engineering, electronic devices, etc. [178]. To formulate self-healing hydrogels, researchers used various techniques with different mechanisms. Covalent and non-covalent bonding mechanisms are involved in the healing mechanism; the covalent bonding mechanism includes the Diels-Alder reaction, imine, boronat, acyl hydrazone, and disulfide types bonding. Noncovalent interactions are ionic interaction, hydrogen bonds, hydrophobic interaction, etc. Those hydrogels formed with these non-covalent types of interactions are flexible and have self-heal properties due to the ability to easily break and reconstruct crosslinks. Hydrogels formed with covalent bonding have good stability [179]. The various self-healing mechanisms of chemical and physical interactions are shown in Figure 5.

Certain approaches are also developed in reservoir modifications, especially in extrinsic self-healing hydrogels, to make them intrinsic and extrinsic self-healing mechanisms. Although there are certain limitations associated with the self-healing properties of some mechanisms, like those that show their effectiveness only at a specific temperature and pH conditions, a lot of research and modifications, e.g., incorporating specific components to make them stimuli-responsive, are ongoing to overcome these issues. Recent advancements have been made with these self-healing hydrogels with excellent properties, including conductivity, adhesiveness, injectability, etc., with broad applicability and specific applicability in certain areas [170]. The self-healing process is demonstrated in Figure 6.

Self-healing hydrogels have a variety of potential applications, as shown in Table 5.

Generally, epidermal injuries heal rapidly, and superficial wound healing occurs through natural mechanisms like epithelialization. In the case of deep wounds, this natural healing mechanism epithelialization does not occur effectively as a destruction systemic host defense mechanism. These are prone to microbial infection and develop non-healing wounds [201,202]. Advancements are initiated to overcome and address these issues in developing wound dressing materials. One of the approaches is self-healing hydrogel which promotes wound closure and healing. These act as multifunctional hydrogels with conductivity, adhesiveness, contractility, etc. [203]. Due to angiogenesis and growth factor deficiency, diabetic wounds are concerned in the healing process and become chronic non-healing wounds. To overcome these, some attempts have been made to improve wound healing in type 2 diabetic induced rats [204]. Another approach by Zhang et al. demonstrated the efficiency of magnetic nanoparticle-based self-healing. One novel and exciting approach of bioactive glass containing copper crosslinked with sodium alginate and PEG diacrylate demonstrated excellent wound sealing potential in diabetic wounds [132,205,206]. Hydrogels with a static crosslinking approach demonstrated their applications in drug delivery for many years [98,207]. Certain limitations associated with this approach are less mechanical strength, uncontrolled release and high water content [208,209,210,211]. Comparatively, self-healing hydrogels as modified versions proved to be promising candidates in drug delivery with controlled release, flexibility, and restoring mechanism that also prevents wear and tear and active ingredients leakage [210]. Despite various advancements in cancer therapy, many obstructions are still there to treat clinically [212,213,214]. Various approaches are reported regarding chemical modifications of natural polymers like alginate, cellulose, etc. They can possess dynamic bonds, which give self-healing properties.

Doxorubicin is a well-explored drug in the self-healing hydrogel approach. Self-healing hydrogels are administered as injectable delivery. In application, due to its shear-thinning property, it is squeezed, and upon reaching the target site, it becomes a gel. These hydrogels are also proven for good stability and damage repair with less administration frequency.

Hydrogel with an inflammation region responsive mechanism loaded with antibiotics and anti-inflammatory agents is also reported as a modified approach. One more approach of injectable hydrogel with responsiveness to pH and ROS reactive oxygen species is also reported to have good antibacterial and anti-inflammatory properties [215]. In the treatment of various diseases and tissue regeneration, cell therapy is proven as a viable approach. Due to the structural similarities with the natural 3D structure of the extracellular matrix, hydrogels have excellent potential in the cell therapy approach. Hydrogels with cell encapsulation promote desired tissue-specific biomechanical and biochemical characteristics that can control cellular functions [216,217]. Properties like biocompatibility and biodegradability are also crucial for applications of hydrogels in cell delivery [218]. In cartilage and bone tissue engineering, self-healing hydrogels have wide applications due to their dynamic and reversible properties. They have a dynamic 3D microenvironment that enables cellular phenotype maintenance and ECM deposition. The desirable mechanical properties, electrical cues, and recovery capacity of self-healing hydrogels play crucial roles in tissue engineering [170]. Overall applications of hydrogels in various areas are as shown in Table 6.

## 7. Patent and Current Clinical Trial Status of the Hydrogel Drug Delivery System

Hydrogels have a long evolutionary history and vast and diverse applications. They are biocompatible, biodegradable, and permeable to gases and nutrients, mechanical strength, and controlled release properties, and they can be tailor-made as they are easily tunable. First-generation hydrogels were simple, highly swellable matrices that involved chemical treatment of monomer/polymer with initiator and crosslinking. Second-generation hydrogels are known as intelligent hydrogels as they respond to external stimuli like changes in temperature, magnetic field, pH, ions, radiation etc. Thermoresponsive hydrogels are intelligent as they show sol to gel phase transition upon a change in temperature conditions [65,233,234]. Some of the polymers that can exhibit thermoresponsive characteristics are poloxamers (pluronic F127 (PF127)), poly(organophosphazenes); poly(*N*-isopropyl acrylamide) (pNIPAAm) and polyoxazoline from synthetic origin and collagen, carrageenans, gelatin, chitosan, starch, xanthan gum, dextran, hyaluronic acid etc. from the natural origin [140,182]. Due to their versatile nature, they are widely explored for various biomedical applications. This is evident from the patents mentioned in Table 7. Due to their vast array of applicability, various products are under clinical trials (Table 8), and many are commercialized successfully.

The popularity of intelligent/intelligent hydrogels that respond to external stimuli has gained immense interest in fields like tissue engineering, drug delivery, cosmetics, agriculture, hygiene products, biosensors, etc. As mentioned below, the advanced properties of thermoresponsive materials have increased stability and efficiency, increasing FDA-approved and commercial products.

Aquatrix™ II, cosmetic hydrogels by Hydromer made with a combination of Polyvinyl Pyrrolidone and chitosan or PVP and Polyethylenimine. These water-based solutions undergo sol-gel transformations, depending on gel strength, within seconds to minutes. These are biocompatible and can hydrate the skin and give a protective, smooth and nongreasy feel. It also has a conditioning effect on hair. It can act as a base for many active water-soluble ingredients [267].

Lip patch ™ by Taiki is a lip shape hydrogel mask for keeping lips elastic and hydrated. It forms a highly nonpermeable layer on lips reducing water loss and showing moisturizing and occlusion effect. It is composed of tocopherol, which has antioxidant properties. Other ingredients are glycerin, sodium hyaluronate, and berries complex [268].

Reversible sol-gel phase transition upon a temperature change is the key feature of Mebiol^®^ gel. It is used for cell implantation, spheroid culture, drug delivery applications, and organ and tissue regeneration as it is liquid when cooled so that cells can be seeded easily. When it undergoes a phase transition to gel, it provides a supportive environment for cell migration, proliferation, protection, and the exchange of gases and nutrients. Mebiol^®^ is composed of poly(*N*-isopropyl acrylamide) and poly(ethylene glycol) (PNIPAAm-PEG) [269].

Avastin(^®^)/PLGA-PEG-PLGA hydrogels are porous, thermoreversible, intravitreal, injectable hydrogels for sustained release of Avastin^®^ for 14 days in-vitro. These systems are safe with no toxicity against retinal tissue. Combining PLGA-PEG-PLGA to Avastin^®^ shifted sol-gel transition to low temperature. These are used for treating posterior segment disease of the eyes [270,271].

Purilon gel by Coloplast provides a moist environment at the wound site. It can be used along with secondary dressing for sloughy wounds, non-infected diabetic foot ulcers, and first and second-degree burns. Its unique hydrating and absorbing properties support autolytic debridement, and once it is applied, it remains at the affected site for a long time. It can be removed easily even after absorption of wound exudates. Purilon gel is composed of sodium carboxymethylcellulose and calcium alginate [272].

iNTRASITE Gel by Smith Nephew can be used in all the stages of wound healing. It contains propylene glycol and is amorphous and in a partially hydrated form, which helps in providing moisture to the drier areas (necrotic tissue) and absorbing exudates from the wet areas of the wound. It is nonadherent, i.e., easy to remove without disturbing viable tissue. It uses a unique Applipak system for application to areas that are difficult to access [273].

LeGoo is used for temporary endovascular occlusion of blood vessels and is composed of thermoreversible, safe, biocompatible poloxamer 407. It undergoes a phase transition to a self-forming plug occupying space in the blood vessel and preventing blood flow once injected into the body. This product can be sold only on the order of a physician according to US Federal Law [274].

Gantrez™ polymers by Ashland is used as bioadhesive in oral care product, used as the base polymer for making polymer salts used as a denture adhesive. It is composed of copolymers of methyl vinyl ether and maleic anhydride [275].

OncoGel (Paclitaxel in ReGel™) is used for the controlled and targeted release of the anticancer drug paclitaxel, preventing its entry into the blood circulation, and sparing normal cells. Thus, it provides efficient and local drug delivery, limiting the dose-related toxicity of paclitaxel [276].

BST-CarGel^®^ by Piramal Life Sciences, Bio-Orthopaedics Division is indicated to improve cartilage repair and stabilize the microfracture-based blood clot. It is a chitosan-based polymeric scaffold dispersed throughout whole blood, and this mixture is implanted over marrow access holes in lesions of the cartilage [269,277].

## 8. Future Scope

Temperature responsive polymers are upcoming versatile polymers explored for a biomedical application utilizing the significance of physiological conditions (body temperature). Conventional Free radical polymerization method shifted to modern polymerization ATRP and RAFT based methods with electrochemical polymerization. The main crucial factor is to lower LCST in the aqueous phase. Tailored polymer processing for in-vivo tissue engineering applications is challenging because phase behaviors represent an important factor for temperature responsive polymer design. We need to understand factors that influence polymer solubilities, such as pH change, co-solutes, co-solvents composition, and surfactant. Turbidimetry is not ideal for estimating aggregation due to dehydration and is not suitable for dilute solutions. This issue needs to be overcome carefully with dynamic light scattering, differential scanning calorimetry, and pressure perturbation calorimetry. Interactions of polymer–polymer, polymer–water, and polymer–cosolvents need to be understood carefully for designing ideal temperature-responsive polymers with zero defects when employed for biomedical applications. The transition temperature is a key factor for biomedical applications. Moreover, the compatibility and cytotoxicity of the materials utilized are often the first concerns in such biomedical applications. Not only should the hydrogels be compatible for medical purposes, but the by-products should also be considered and thoroughly analyzed. The production of synthetic bioinspired substances or organic products from high throughput preparation, such as hyaluronic acid, chitosan, gelatin, collagen, and so on, could improve material safety at a cheaper cost. Temperature responsive polymer-based soft robotics, microfluidic delivery, and 3D printing technology need to be designed with a safe, non-toxic, and biocompatible polymer. 

## 9. Conclusions

Research in thermoresponsive polymers for drug and gene delivery, tissue engineering, and wound healing has been well recognized in the past years. Being a class of smart materials, stimuli-responsive hydrogels have the potential to provide targeted drug delivery. Uneven capillary networks, elevated interstitial pressure, and the diffusion of intended stimuli across the tumor cells, on the other hand, could result in undesirable medication leakage and possibly off-target effects during transit. Furthermore, improper drug deposition in solid tumors might lead to drug resistance and significantly reduce the therapeutic impact. Drug-loaded hydrogel capsules that respond to indicators of interest could help improve this research. Hydrogel capsules with increased specificity and penetration offer a new perspective on cancer treatment by modifying aptamers that focus on specific cells. Furthermore, by adopting rapidly developing dynamic covalent bond systems for the construction of multi-functional hydrogels, the rigidity and low stability difficulties of the conventional hydrogel system may be addressed, and this will undoubtedly contribute to developing this worthwhile subject. Currently, thermogel has shown very high potential as a drug delivery option for several diseases, such as cancer, HIV infection, oral infections, wound healing, tissue engineering, and gene therapy. It is an excellent choice because of the universality of tapping on physiological temperatures for biomedical applications. The modified thermogel is highly effective in several drug delivery technologies, and most importantly, all these systems are biodegradable and very safe, thus presenting them as a better choice. The advent of supra-molecular hydrogels will encourage us to look beyond the depot forming property of thermogels and discover other use cases that tap upon the dynamic and reversible nature of thermogelling polymers.

## Figures and Tables

**Figure 1 polymers-14-03126-f001:**
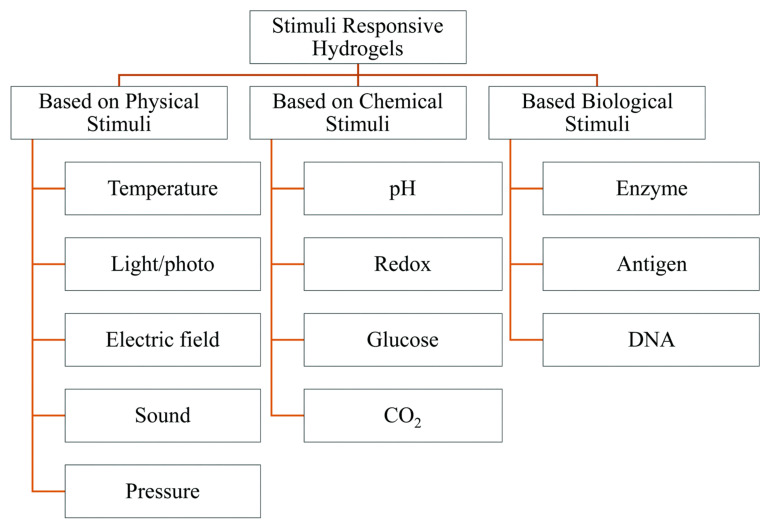
Classification of stimuli-responsive hydrogel based on physical, chemical, and biological stimuli. Reproduced with permission from Sikdar et al. [3], ©The Royal Society of Chemistry, 2021.

**Figure 2 polymers-14-03126-f002:**
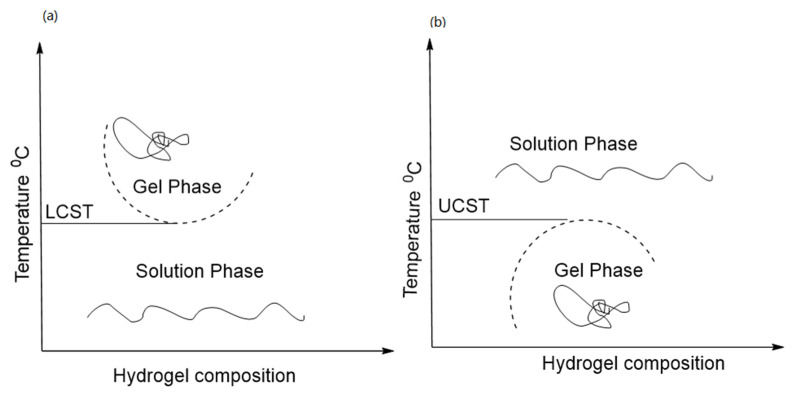
(**a**) the lower critical solution temperature (LCST) formed hydrogel on increasing temperature, and (**b**) the upper critical solution temperature (UCST) formed hydrogel on cooling.

**Figure 3 polymers-14-03126-f003:**
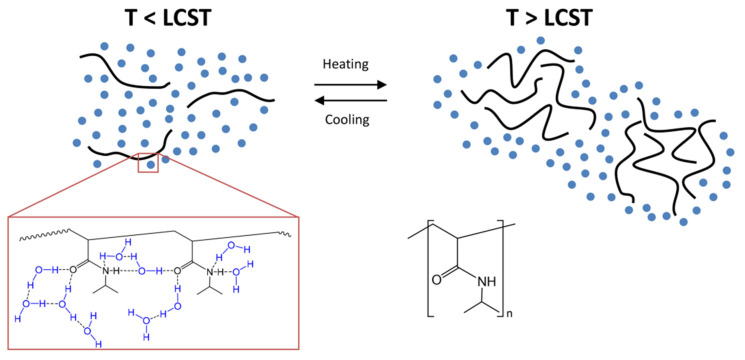
PNIPAAm chains (black) surrounded by water molecules (blue) as a function of temperature. Bottom-right: chemical structure of PNIPAAm. The red inset shows the possible hydrogen bonds between water molecules and polymer chains. Below the LCST, polymer chains are fully hydrated and solubilized, whereas above the LCST, they interact strongly with one another, the intrachain hydrophobic effect changes the conformation of the polymer chains to a coil state, they aggregate, and phase separate from the water phase to yield a turbid suspension. Reproduced from Bordat et al. [94] with kind permission of the copyright holder, 2019, Elsevier.

**Figure 4 polymers-14-03126-f004:**
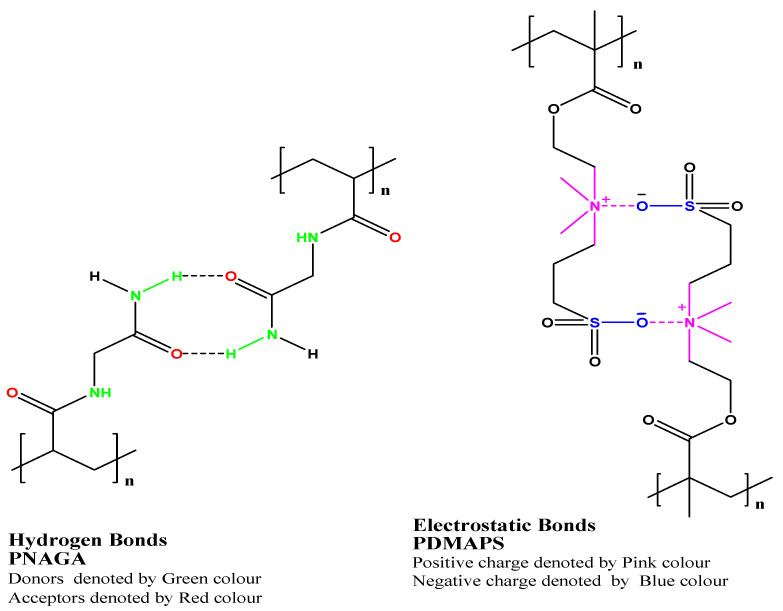
Thermoresponsive behaviors in the UCST copolymers are achieved by hydrogen bonds between poly(*N*-acryloyl glycinamide) polymer chains and electrostatic bonds between zwitterionic groups in poly(*N*, *N*′-dimethyl(methacryloylethyl)ammonium propane-sulfonate). Reproduced from Niskanen et al. [44], with the kind permission of the copyright holder, Royal Society of Chemistry, an open access article distributed under the Creative Commons Attribution License that permits unrestricted use, distribution, and reproduction in any medium.

**Figure 5 polymers-14-03126-f005:**
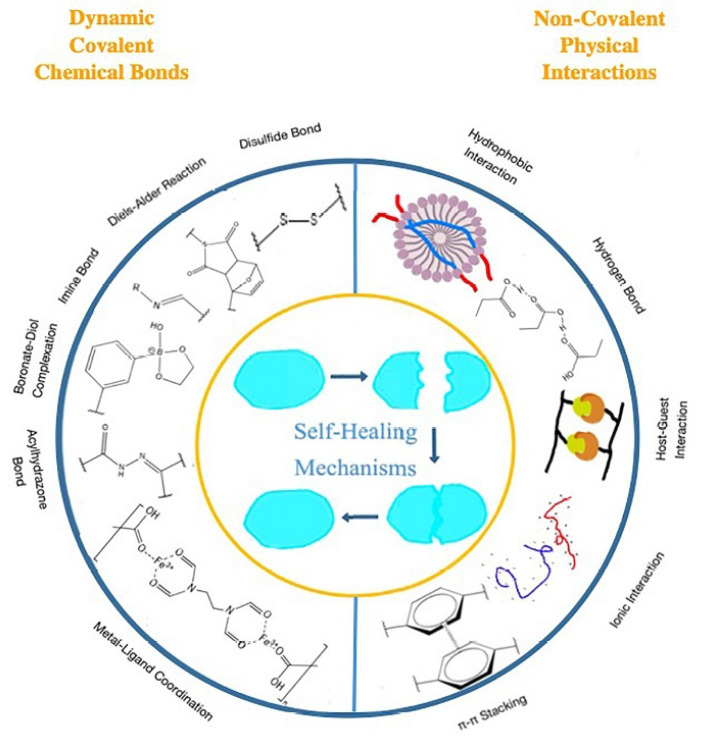
Self-healing mechanisms: Chemical covalent bonds and physical non-covalent interactions. Reproduced from Fan et al. [178] with kind permission of the copyright holder, 2020, Frontiers, an open access article distributed under the Creative Commons Attribution License that permits unrestricted use, distribution, and reproduction in any medium.

**Figure 6 polymers-14-03126-f006:**
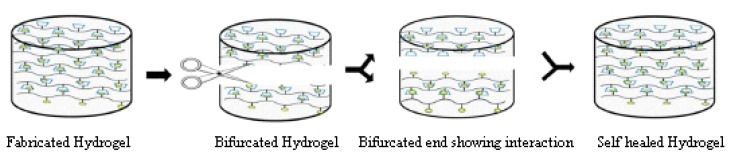
Demonstration of the healing process in a hydrogel [35].

**Table 1 polymers-14-03126-t001:** Drug release exponent values (*n*) in the empirical power-law model proposed by Peppas et al., adapted from [97,104]. Reproduced from Lin et al. [97], with kind permission of the copyright holder, 2006, Elsevier; Reproduced from Siepmann and Pepaas [104] with kind permission of the copyright holder, 2001, Elsevier.

Matrix/Geometry Type	Diffusion-Controlled Drug Delivery System (Case I)	Swelling-Controlled Drug Delivery System (Case II)
Slab	*n* = 0.5	*n* = 1
Cylinder	*n* = 0.45	*n* = 0.89
Sphere	*n* = 0.43	*n* = 0.85

**Table 2 polymers-14-03126-t002:** The kinetic and reaction controlled hydrogel, their subtypes and drug release mechanism.

System Type	Subtype	Mechanism	Example	References
**Kinetically controlled systems**	Pendant chain	The drug is covalently bound to the hydrogel through breakable spacers, and the rate of the spacer-bond breakage controls drug release	Fibrin matrix bounded with pendant VEGF factors variants attached through the plasmin sensitive peptidyl substrates	[125]
	Surface eroding	Drug release is facilitated through surface erosion of polymer matrices	In-vitro enzymatic surface erosion of the degradable poly (ethylene glycol)-poly-caprolactone (PCL-b-PEG-b-PCL) block-copolymer hydrogel systems in the presence of a high concentration of the lipase	[126]
**Reaction controlled systems**	Bulk degrading	The drug release profile is facilitated through both network degradation (chemical reaction) and drug diffusion	PEG–PLA block co-polymers could be polymerized to form hydrolytically degradable hydrogel drug delivery systems	[127]
	Affinity type	Reaction type reversible drug release mechanism works on the principle of affinity of the hydrogel systems, mainly used for therapeutic proteins-ligand delivery	The Heparin-loaded hydrogel matrix controlled the release rate of these growth factors by affinity binding.	[128,129]

**Table 3 polymers-14-03126-t003:** Miscellaneous types of hydrogel systems and their drug release mechanism.

System Type	Subtype	Mechanism	Example	References
**Dynamic hydrogel drug delivery systems**	Degradable	The drug release rate depends upon the matrix swelling, degradation and the diffusion	In-vitro enzymatic surface erosion or degradation of hydrogel systems and polymer-based microneedle systems or plasma coated drug delivery systems	[97,130,131,132,133]
	Stimuli-sensitive	Drug release is controlled by external stimuli such as temperature, pH and enzymes.	pH-responsive poly (methacrylic acid) (PMAA) hydrogel system for delivery application	[134]
**Composite hydrogel drug delivery systems**	Multi-layer type	The different layers were formed as per the release requirement; at a time, multiple drugs could be released or if required release of a single drug or molecule could be tuned	The multi-laminated hydrogel system developed through the photo-polymerization for multiple protein drug delivery	[135]
	Multi-phase type	The drug release could be controlled by the multi-phase systems, such as the microsphere system in the hydrogel system (several viscosities) for multiple drug deliveries of biologics	Multiple protein drug delivery using the protein loaded microsphere and other protein-loaded hydrogel systems, the microsphere could be placed in the different viscosity of the hydrogel to tunned the drug release from the two phases. E.g., protein-loaded PLGA microspheres in hydrogel	[136]
**Micro/nano-scaled hydrogel drug delivery systems**	This hydrogel system is prepared from the hydrophilic polymer. Generally, nano or microparticles were developed and loaded in the gel for single or multiple drug delivery; the type of polymer could control the drug release of the drug. And drug release could be predicted using diffusion or the monte Carlo model.		Protein-loaded PLGA microspheres in hydrogel	[137]
**In-situ forming hydrogels drug delivery systems**	Drug release could depend upon the monomer/polymer used with different functionalities in this system. The solution form is converted into the gel form in-vivo, which regulates the drug release from the matrix; this could be based on the temperature or pH.		In-situ hydrogel-based delivery of the proteins, peptides	[138]

**Table 4 polymers-14-03126-t004:** Thermoresponsive hydrogel containing biotherapeutics for the treatment of various diseases.

Polymeric Carriers	Encapsulant	Gelling Temp	Comments	Ref.
Poloxamer 407, Poloxamer 188 and carbomer 974P	Paclitaxel (PTX)	31–35 °C	Hydrogel has the adequate viscoelasticity and self-recovery. In vivo studies revealed that a PTX-nanocrystal laden gel suppressed both local and distant tumor growth.	[170]
Pluronic F127 and *N*, *N*, *N*-trimethyl chitosan	Docetaxel (DTX)	30–35 °C	Pure DTX and DTX loaded PF127 hydrogel are less efficient at killing U87MG cells than DTX loaded PF127-TMC hydrogel.	[34]
Chitosan/hyaluronic acid/β-sodium glycerophosphate (CS/HA/GP)	Doxorubicin (DOX)	31.2–37.2 °C	With increasing HA concentration, the gelling temperature of CS/HA/GP steadily declines and falls.	[171]
Chitosan/β-sodium glycerophosphate/polyethylene glycol (CGD)	Doxorubicin	31–35 °C	Due to the development of Schiff base bonds among the amino groups in chitosan and the aldehyde groups in PEG, DOX-loaded CGD hydrogels had lower gelling temperatures and higher viscosity.	[172]
D-PNAx nanomedicines	Doxorubicin	34–44 °C	Thermoresponsive sol-gel phase transitions of D-PNA100 nanoparticles observed in the range of 5.0 to 10.0% of D-PNA100 concentration, with CGTs decreasing from 38 °C at 5.0% to 32 °C at 10.0% as concentration rises.	[173]
Levan/*N*-isopropyl acrylamide	5-aminosalicylic acid	32.8–35.09 °C	The concentration of levan positively influenced the biocompatibility of the hydrogels. Moreover, when the amount of levan in the hydrogels increased, so did the amount of levan on the hydrogel surface.	[174]
Poly(ethylene glycol)-poly(sulfamethazine carbonate urethane	Lysozyme	37 °C	Following subcutaneous administration in SD rats, lysozyme-loaded PEG-PSMCU composites produced an in-situ hydrogel, which significantly delayed the first burst and resulted in lysozyme release that lasted for 7 days.	[175]
Chitosan/b-glycerophosphate/collagen	Human adipose tissue-derived stem cells (ADSCs)	36–38 °C	The capacity of ADSCs embedded hydrogel to develop into fatty tissue was also demonstrated in an in vivo investigation, indicating high histocompatibility and good adipogenesis potential.	[176]
Ferrimagnetic chitosan hydrogel (FCH)	Iron oxide Nanocubes (IONCs)	37 °C	DOX-loaded ferrimagnetic chitosan hydrogel had a synergistic impact and provided long-term treatment for tumor cells.	[177]

**Table 5 polymers-14-03126-t005:** Self-healing hydrogels potential applications.

Hydrogel Applications	References
Tissue engineering	[178,179,180,181]
Drug delivery	[178,179,182,183,184,185,186,187]
Wound management/healing	[131,178,179,188,189,190,191,192,193]
Miscellaneous applications	[35,51,52,53,54,55,170,171,186,187,188,189,190,191,192,193,194,195,196,197,198,199,200]

**Table 6 polymers-14-03126-t006:** Hydrogel applications overview.

Types of Hydrogel	Potential Application	References
Thermo-responsive	Tissue/Skin regeneration, wound healing,	[163,219,220]
Photoresponsive	Delivery of drugs, micro-fluidic devices	[163,221,222]
Electro responsive	Implant drug delivery	[178,223,224]
Magnetic responsive	Tissue repair, Diagnosis and targeting, Drug delivery,	[178,225,226,227,228,229]
pH-responsive	Protein and drug delivery, 3D cell culture	[178,221,222,230]
Glucose responsive	Devices, Immuno-isolation	[178,231,232]

**Table 7 polymers-14-03126-t007:** Exploring the potential of thermosensitive hydrogels through patents.

Sr. No	Patent Number and Year	Title	Proposed Use	Findings of Invention/Summary	Inventors
1	US 20210361826, 2015 [235]	Biodegradable, Thermally Responsive Injectable Hydrogel for Treatment of Ischemic Cardiomyopathy	Ischemic Cardiomyopathy	Method of preparation and applications of biodegradable, thermoresponsive, elastomeric Material, especially copolymers of NIPAAm—*N*-isopropyl acrylamide (NIPAAm), *N*-vinylpyrrolidone and methacrylate-polylactide macromer residues are described. These have an LCST of less than 37ᴼC and degradation rate < 200 days in vivo. These compositions can be used for treating defects in heart muscle.	Hongbin Jiang, William R. Wagner, Tomo Yoshizumi, Yang Zhu
2	*US20140065226A1, 2011* [236]	Thermo-responsive hydrogel compositions	Drug delivery for wound healing or Hydrogel loaded with Nanospheres for Ocular Application	The patent discloses the composition of thermoresponsive hydrogel synthesized by Radical polymerization, consisting of an acrylamide crosslinked with PEG -diacrylate and monomer containing amino acid. This thermoresponsive hydrogel shows a dual change in physicochemical characteristics when it comes in contact with the body temperature of mammal and releases embedded drug in a controlled manner	Eric Brey Jennifer J. Kang-Mieler, Victor Perez-Luna, Bin Jiang, Pawel Drapala, Rolf Schäfer, Hans Hitz
3	WO2014138085A1, 2014 [237]	The thermoresponsive hydrogel containing polymer microparticles for noninvasive ocular drug delivery	Ocular drug delivery	Self-administrable thermoresponsive hydrogel for ocular delivery of bioactive is discussed. The hydrogel consists of an elastin-like peptide, a polysaccharide. The drug is entrapped in polymeric microparticles, further embedded in the thermoresponsive hydrogel.	Morgan V. Fedorchak, Steven R. Little Joel S. Schuman Anthony Cugini
4	WO2019092049A1 WIPO (PCT) 2018 [238]	A thermo-responsive hydrogel for intratumoral administration as a treatment in solid tumor cancers	Solid tumors	A thermosensitive hydrogel that can be injected is formed using 15–25% poloxamer polymer along with chitosan, 2-Hydroxypropyl β-cyclodextrin and genipin. This hydrogel can be used to incorporate chemotherapeutic agents for treating solid tumors.	Helena Kelly, Garry Duffy, Seona Rossi, Conn Hastings
5	US20070116765A1, 2004 [239]	The aqueous dispersion of hydrogel nanoparticles with inverse thermoreversible gelation	Controlled drug delivery	Hydrogel nanoparticles have an interpenetrating polymer network with inverse thermogelation properties for drug delivery applications. An aqueous dispersion of hydrogel nanoparticles can release the drug in a time-dependent manner. Polymers used for preparation are poly(*N*-isopropyl acrylamide), and monomer comprises acrylic acid along with cross-linking agents such as N, N′-methylenebisacrylamide or N, N′-methylenebisacrylamide; potassium persulfate; ammonium persulfate are used as initiators; sodium dodecyl sulfate is used as a surfactant.	Zhibing Hu, Xiaohu Xia
6	US20170296672A1, 2015 [240]	Non-ionic and thermoresponsive diblock co-polypeptide hydrogels for delivery of molecules and cells	Delivery of drugs or cells and injecting cells into CNS.	The composition of co-polypeptide thermoresponsive hydrogel for delivery of the pharmaceutical substance, nucleic acid, peptide, hormone, or imaging agent is disclosed. The hydrogels are synthesized using a hydrophilic segment of poly methoxy ethoxy-ethyl-rac-glutamate for preparing nonionic diblock co-polypeptide hydrogels	Timothy J. Deming, Michael V. Sofroniew, Shanshan Zhang
7	US-8858998-B2, 2008 [241]	Thermoresponsive Arginine-based Hydrogels as Biologic Carriers	Biomedical applications for drug delivery	Cationic poly (ester amide) (PEA)-based hydrogels are fabricated using precursors such as unsaturated L-arginine based poly (ester amide) (UArg-PEA), pluronic DA or a combination. Hydrogels based on Pluronic DA/UArg-PEA combination and pure pluronicDA were thermosensitive, but pure UArg-PEA-based hydrogels were only biodegradable but not biodegradable thermoresponsive. These synthesized hydrogels can be utilized for various biomedical applications, especially drug delivery.	Chih-Chang Chu Hua Song
8	EP3708167A1, 2017 [242]	Immunomodulating treatments of body cavities	Cancer therapy	Treatment of cancer of internal body cavities (like cancer of the Urinary tract) and thus providing local drug delivery to the inaccessible regions in the body. It can also be used to deliver a combination of controlled drug delivery and immunomodulatory agents	Gil Hakim, Astar Friedman, Marina Konorty, Dalit Strauss-Ayali
9	US20190343761A1, 2017 [243]	Antibiotic formulations for lower back pain	relieve and treat low back pain	Discloses composition, methods of preparation of injectable, thermosensitive hydrogel containing a radio-contrast agent, a drug belonging to an antibiotic class, used for easing lower back pain	Lloyd Czaplewski, Sarah Guest
10	US20190030211A1, 2018 [244]	Hydrogel scaffold for three-dimensional cell culture	It encapsulates the cells in a 3D hydrogel scaffold that forms the engineered tissue. Methods of making engineered tissues.	This invention discusses the preparation and composition of an electrospun microfiber scaffold based on a combination of thermoresponsive polymer and biodegradable polymer for encapsulating cells for making engineered tissues. Thermoresponsive polymers (PEG)-poly(*N*-isopropyl acrylamide) and biodegradable polymers like PCL are mixed in the ratio of 65:35	Jin Nam, Alexander Brunelle
11	US20190336648A1, 2017 [245]	Bone-promoting thermoresponsive macromolecules	Bone formation/repair and the treatment of bone diseases.	The thermoresponsive hydrogel formed via carbodiimide chemistry between peptide group covalently linked with the carboxyl group of citric acid monomers. The peptide is cyclic Arg-Gly-Asp (cRGD) which is conjugated covalently to carboxy groups of (Polyethyleneglycol citrate-co-*N*-isopropyl acrylamide) (*PPCN*). These are used for the delivery of bioactive agents.	Guillermo A. Ameer Simona Morochnik
12	US20210205459A1, 2019 [246]	Injectable thermoresponsive hydrogels as a combinatory modality for controlled drug delivery, biomaterial implant and 3d printing bio link	Drug delivery, implants, 3D printing bio link	Mechanical Stiffness and strength of Insitu thermoresponsive polymeric hydrogels formed using Polyethylene glycol, hyaluronic acid, polyvinyl chloride or methylcellulose were improved using cellulose derivatives such as Cellulose nanofibers/crystals. This combination can be used to control drug release or as an implant and bio ink for 3D printing and treating bone disorders, preventing cancer/infectious diseases.	Soumya Rahima Benhabbour, Panita Maturavongsadit
13	US20200100931A1 [247]	Thermoresponsive Skin Barrier Appliances	Wound healing	The patent discloses thermoresponsive ostomy (body wastes discharged through a surgically created opening in the body) and skin barrier appliances for wound healing. It describes an assembly consisting of a pump that expels biosealant to the pump output port upon being stimulated. It causes the collection of hydrogel beads (composed of NIPAAm) to vibrate with high energy. This friction causes localized heating, leading to the bead plug layer contracting or swelling in size. It can also sense wound leakage and absorb wound exudate	Jeffrey Norman, Schoess Kannan, Sivaprakasam
14	US2021317267A1, 2021 [248]	Thermogelling supramolecular sponge as self-healing and biocompatible hydrogel	Carrier materials for active ingredients such as drugs, cells, proteins and bioinks for 3D bioprinting in tissue engineering	Synthesis of block copolymers made up of poly (2-oxazine) and poly (2-oxazoline) is discussed. These hydrogels have advanced and efficient rheological and thermoresponsive characteristics due to specific structures [A].sub.n-[B].sub.m or [B].sub. N-[A].sub.m, where n and m have the approximately same value and range from 20 to 300.	Lorson, Thomas Luxenhofer, Robert
15	US 20210106708, 2019 [249]	THERMORESPONSIVE COMPOSITIONS AND METHODS FOR PREVENTING AND DISRUPTING BIOFILMS	Medical implant coated/impregnated with nanocomposite for disrupting or preventing biofilm formation.	A medical implant that is resistant to biofilm formation, wherein the medical implant is at least partially coated or impregnated with the nanocomposite A thermosensitive polymeric nanocomposite composed of one or more D-amino acids and one or more energy-actuatable particles is discussed. When the energy source is excited, it cause localized heat release from the nanocomposite leading to sol to gel transition of a glycol chitin-based hydrogel.	Anna Cristina S. Samias Alvatore J. Frangiamore Carlos A. Higuera Ruedaalison K. Klikawael K. Barsoum
16	WO2014138085A1, 2014 [237]	The thermoresponsive hydrogel containing polymer microparticles for noninvasive ocular drug delivery	Ocular drug delivery	This patent discloses the formulation method of drug-loaded polymeric microparticles embedded in thermoresponsive hydrogel for topical delivery to the ocular surface for treating glaucoma, conjunctivitis, chronic dry eye etc. Polymeric microparticles were composed of dextran, PLGA, PLA, PCL, alginate etc. Hydrogel comprises polyacrylamide, a silicon hydrogel, PEO/PPO, polyacrylic acid, N, N′-dimethyl aminoethyl methacrylate, which sustained release for up to 30 days.	Morgan V. Fedorchak Steven R. Little Joel S. Schuman Anthony Cugini
17	US10767037B2, 2016 [250]	Hyaluronic acid conjugates and uses thereof	Tissue engineering, cosmetics, drug delivery	Self-lubricating nano-ball-bearing (SLNBB) properties of Hyaluronic acid and *N*-isopropyl acrylamide-based polymer graft polymers are explored. Injectable, biocompatible, stable, biodegradable pH and thermo-sensitive polymeric hydrogels with long residence time at the injection site due to the formation of the spontaneous nanoparticles. These can be used as viscosupplementation/lubrication material for drug delivery or cosmetic applications	Pierre Maudens, Eric Allemann, Olivier Jordan
18	US9937254B2, 2011 [251]	Water-soluble supramolecular complexes	Solid dosage form for pharmaceutical, diagnosis or cosmetic use.	The water solubility of drugs can be improved when formulated as water-soluble supramolecular hydrogel complexes that form a transparent, thermoreversible gel upon the combination with water. They may be hydrated or dehydrated repeatedly for insoluble drugs. These are composed of at least two blocks of polyethylene oxide and at least one block of polypropylene oxide.	Shao Xiang, LuJeffrey LuLetian Liu
19	US20100098762A1, 2008 [252]	Thermosensitive Pluronic Derivative Hydrogels With High Biodegradability and Biocompatibility for Tissue Regeneration and Preparation Method Thereof	Tissue and organ regeneration	Pluronic-based thermoresponsive smart hydrogels are synthesized for tissue engineering applications. Pluronic is derivatized by conjugating it with biodegradable polymers. The drug/active ingredient is conjugated with methacryloxyethyl trimellitic acid anhydride that is conjugated to the biodegradable polymer	Dong Keun Han, Kwideok ParkJae-Jin Kim
20	US20150266986A1, 2014 [253]	Multifunctional Hyperbranched Polymers	Biomedical applications- wound healing	RAFT (Reverse Addition-Fragmentation chain Transfer) polymerization technique synthesizes PEG-based hyperbranched copolymer. These can be used for delivering antimicrobial agents. These hydrogels are stable for one year, as seen from stability studies. RAFT agents can be Dithiobenzoates, Trithiocarbonates and Dithiocarbamates.	Wenxin WangRobert KennedySean McMahon
21	Indian patent 279339, 2017 [254]	“Injectable hydrogel-forming chitosan mixtures”	Biomedical applications	Aqueous solutions containing chitosan derivatives are synthesized, showing dual responsive behavior, i.e., temperature sensitivity and pH-dependent change in physicochemical characteristics. These can be utilized for various biomedical applications.	Ben-Shalom Noah, Nevo Zvi, Patchornik Avraham, Robinson Dror
22	US20120231072A1, 2012 [255]	Thermo-responsive hydrogel compositions	Wound healing, anti-microbial effect through drug or drug-loaded nanoparticle.	The synthesis method of smart, thermo-responsive hydrogel consists of monomer and polymer having an amino acid side chain (comprises an amino acid linked to an acrylic-, maleic-, or phthalic-derivative or *N*-isopropyl acrylamide).	Jennifer J. KANG-Mielereric Breyvictor PEREZ-Lunabin Jiangpawel Drapalahans Hitzrolf Schaefer
23	US20090053276A1, 2008 [256]	Injectable hydrogel compositions	Drug delivery	Thermosensitive hydrogels in dry form or hydrated form are synthesized in this invention. These injectable hydrogels swell in-vivo in the body because their UCST is below body temperature or their LCST is above average body temperature, i.e., these hydrogels contract when cooled below UCST and expand when heated.	Robert E. Richard
24	US7658947B2, 2010 [257]	Thermogelling composition patent.	Drug delivery	Thermoresponsive hydrogel consisting of methylcellulose and citric acid is described. The developed hydrogel can be utilized for diverse applications like drug delivery, cosmetics, adjuvants, and nutritional agents. Controlled release of pharmaceutical agents through body cavities, topically or subcutaneous injections are possible.	Yanbing, H. Thermogelling composition
25	US20120020932A1, 2012 [258]	Thermosensitive hydrogel composition and method patent.	Drug delivery	Drug-loaded injectable thermosensitive hydrogel composed of methylcellulose as the thermoresponsive polymer is synthesized. It also contains extracellular matrix protein and Hyaluronic acid. This can remain as a liquid at room temperature for ease of administration and gels; once it reaches the desired site in the body, it sets as a gel due to a change in the temperature.	Jian, Q.Y.
26	US20100098762A1, 2010 [259]	Thermosensitive pluronic derivative hydrogel with high biodegradability and biocompatibility for tissue regeneration and preparation method thereof	Tissue engineering	Biocompatible, thermosensitive and biodegradable hydrogels are synthesized using derivatization of Pluronic. Active constituents are conjugated with derivatized pluronic and utilized for tissue regeneration in tissue engineering	Dong, K.H.

**Table 8 polymers-14-03126-t008:** Data on Clinical trials for thermoresponsive hydrogels.

Status of Clinical Trial	Outcome of Study	Use (Disease and Formulation)	Clinical Trial Identifier
Completed Phase II	Nonsurgical, local, adjunctive therapy for periodontitis treatment using Nitazoxanide loaded into thermoresponsive hydrogels.	Nitazoxanide hydrogel for periodontitis	ClinicalTrials.gov Identifier: Identifier: NCT04768530, 24 February 2021 [260]
Phase I	Hydrogel patch developed for S-flurbiprofen and its bioavailability is compared with the marketed tablet formulation	Flurbiprofen (Nonsteroidal anti-inflammatory drug) hydrogel patch for arthritis or dental pain.	ClinicalTrials.gov Identifier: NCT04505787, 10 August 2020 [261]
NA	Hydrogel based wound dressing for treating Diabetic Foot Wounds is formulated and evaluated, and its efficacy is checked against traditional wound dressing	Hydrogel/nano silver-based dressing for diabetic foot ulcers.	ClinicalTrials.gov Identifier: NCT04834245, 8 April 2021 [262]
Phase 4	Metronidazole hydrogels are developed for sublingual administration to treat periodontitis in Stages I and II	Metronidazole hydrogels for periodontitis	NCT04983849, 30 July 2021 [263]
Phase 4	Bulkamid is synthesized using polyacrylamide hydrogel as a transanal injection for the treatment of anal incontinence	Bulkamid for anal incontinence using transanal injection	ClinicalTrials.gov Identifier: NCT02550899, 12 January 2016 [264]
NA	The safety and efficacy of HEC-hydroxyethyl cellulose hydrogel (PROMGEL-OA) are studied to treat knee pain caused by osteoarthritis.	(PROMGEL-OA) Hydrogel injection for Osteoarthritis	NCT04061733, 4 May 2022 [265]
NA	Local injection for correction of nasolabial folds containing Hyaluronic Acid and Lidocaine	Hydrogel injection for nasolabial folds	ClinicalTrials.gov Identifier: NCT05252325, 23 February 2022 [266]

## Data Availability

Not applicable.

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
