# Peer review of "Functional Thermoresponsive Hydrogel Molecule to Material Design for Biomedical Applications"

_polymers, 2022, doi:10.3390/polym14153126_

Round 1

Reviewer 1 Report

The review manuscript focuses on the thermosensitive hydrogel and its synthesis methods, forming mechanisms, mathematical models for drug release, applications,  and patent. The authors are able to centralize and summarize a large number of important references, also the examples shown in the review are representative. This manuscript can help researchers quickly have a deep understanding about thermoresponsive hydrogel to make further investigation and develop new applications. But some mechanisms and the properties of thermoresponsive hydrogel need to be further explained. I support the publication after the careful revision by the authors throughout the manuscript. Below are specific comments:

1.The author could also summarized the mechanical strength, adhesion property, optical properties, and lasting time of several typical thermoresponsive hydrogels to further introduce it in the introduction part.

2.Chaper 2 is the total score structure, the authors mention the natural polymers and their derivatives then followed with examples classified as polysaccharides, proteins, N-Isopropylacrylamide(pNiPAAm)-based systems, PEO/PPO-based systems, PEG/Biodegradable polyester copolymers and Poly(organophosphazenes). But the synthesis polymers are ignored. Could the authors systematically reveal the application of synthesis polymers with representative references?

3.Part 3.2 to 3.5 are synthesis method of hydrogel sorted by production method while part 3.1 and 3.6 to 3.8 by chemical mechanism, they are not parataxis. Please rebuild the structure of this chapter to make the manuscript more logically.

4.In chapter 4 when the authors reveal the mechanism of the phase transition in thermoresponsive hydrogel, they explain the influence of temperature to hydrogen, but the mechanism can be further explained by the change of entropy and Gibbs free energy as other previous studies mention.

5.The authors can add predictive applications in the future scope part.

6.In line 76, UCST is the abbreviation of upper critical solution temperature. The word upper is missing. In line 80, the full name of NIPAAm can be mentioned to make readers more clearly. In line 461, please change tune into turn. In line 495, please change 4.3 into 4.5. In line 717, please change 13-15 into 13-14.

7.The authors should carefully recheck the reference number in the passage, some numbers do not match the correct reference, such as: line 300 Ref. 43, line 318 Ref. 3, line 392 Ref. 55, line 596 and 607 Ref. 79, line 908 Ref. 166. Please update all of them in the revised version.

8. The authors could consider adding the following articles into references which would again increase the interest to general smart hydrogel readers: Chemical Society Reviews, 2021, 50, 8319-8343; Chemical Society Reviews‚ 2022‚ 51, 4175-4198ï¼›Advanced Functional Materials‚ 2022‚ 32, 2108749.

Author Response

Reply to reviewers’ comments to:

Ms. Ref. No.: polymers-1748682

Title: Functional Thermoresponsive Hydrogel Molecule to Material Design for Biomedical Applications

The authors are thankful to each reviewer for their thorough investigation of the manuscript and the thoughtful comments on the same. The constructive suggestions by the reviewers have certainly improved the understanding of the manuscript from its previous state. The authors have addressed all the comments to the best of their knowledge, and any changes arising from them have been made in the manuscript and highlighted in colors for the reviewers. The changes are also categorically pointed out in the replies to the comments, which are as follows:

Reviewer #1

The review manuscript focuses on the thermosensitive hydrogel and its synthesis methods, forming mechanisms, mathematical models for drug release, applications, and patent. The authors are able to centralize and summarize a large number of important references, also the examples shown in the review are representative. This manuscript can help researchers quickly have a deep understanding about thermoresponsive hydrogel to make further investigation and develop new applications. But some mechanisms and the properties of thermoresponsive hydrogel need to be further explained. I support the publication after the careful revision by the authors throughout the manuscript. Below are specific comments:

  1. The author could also summarized the mechanical strength, adhesion property, optical properties, and lasting time of several typical thermoresponsive hydrogels to further introduce it in the introduction part.

Response: Thank you for the comments as suggested by the reviewer, physicochemical properties of thermoresponsive hydrogel have been included in the introduction part in the main text of final manuscript.

2.Chaper 2 is the total score structure, the authors mention the natural polymers and their derivatives then followed with examples classified as polysaccharides, proteins, N-Isopropylacrylamide(pNiPAAm)-based systems, PEO/PPO-based systems, PEG/Biodegradable polyester copolymers and Poly(organophosphazenes). But the synthesis polymers are ignored. Could the authors systematically reveal the application of synthesis polymers with representative references?

Response: Thank you for the comments as suggested by the reviewer, synthetic polymers are covered from 2.2.3 to 2.2.6. Inadvertently, we missed out to give headline as “synthetic polymers and their derivatives”. Now, we have updated the text.

3.Part 3.2 to 3.5 are synthesis method of hydrogel sorted by production method while part 3.1 and 3.6 to 3.8 by chemical mechanism, they are not parataxis. Please rebuild the structure of this chapter to make the manuscript more logically.

Response: Thank you for the comments as suggested by the reviewer, correction was made and updated in the revised version of the manuscript.

  1. In chapter 4 when the authors reveal the mechanism of the phase transition in thermoresponsive hydrogel, they explain the influence of temperature to hydrogen, but the mechanism can be further explained by the change of entropy and Gibbs free energy as other previous studies mention.

Response: Thank you for the comments as suggested by the reviewer, the entropy and Gibbs free energy concept have been included under the phase transition mechanism.

  1. The authors can add predictive applications in the future scope part.

Response: Thank you for the comments as suggested by the reviewer, the future scope part has been updated in the revised manuscript.

6.In line 76, UCST is the abbreviation of upper critical solution temperature. The word upper is missing. In line 80, the full name of NIPAAm can be mentioned to make readers more clearly. In line 461, please change tune into turn. In line 495, please change 4.3 into 4.5. In line 717, please change 13-15 into 13-14.

Response: Thank you for the comments as suggested by the reviewer, all the typographical mistakes were corrected in the revised version of the manuscript.

7.The authors should carefully recheck the reference number in the passage, some numbers do not match the correct reference, such as: line 300 Ref. 43, line 318 Ref. 3, line 392 Ref. 55, line 596 and 607 Ref. 79, line 908 Ref. 166. Please update all of them in the revised version.

Response: Thank you for the comments as suggested by the reviewer, we have thoroughly checked and revised the references in the final version of the manuscript.

  1. The authors could consider adding the following articles into references which would again increase the interest to general smart hydrogel readers: Chemical Society Reviews, 2021, 50, 8319-8343; Chemical Society Reviews‚2022‚51, 4175-4198ï¼›Advanced Functional Materials‚ 2022‚ 32, 2108749.

Response: Thank you for the comments as suggested by the reviewer, we have fully considered this paper as a potential reference and cited it in the final review version of the manuscript.

Reviewer 2 Report

The review titled as "Functional Thermoresponsive Hydrogel Molecule to Material Design for Biomedical Applications" summarize materials about thermoresponsive hydrogel for biomedical applications. The review can be published in Polymers mdpi after major revision. 

The following questions may be clarified.

In the review is absent information about nanogels and hydrogel coatings.

 Please add chemical structures which were mentioned in the review. 

Information about different gels systems should be summarized in Table.

What means "In radical polymerization, most monomers include hydrophilic  –C C– groups [45]." (line 322-333) It is not clear to me.

In general, subsection 4. "Mechanism of functional thermoresponsive hydrogel" is difficult for understanding and should be revised. Transition for the PNIPAM was described incorrectly (what about the interaction between amide fragments at T above LCST). Information about a phenomenon named as hydrophobic hydration is absent.

Information about the impact of the buffer solution or body fluid on the temperature-responsive properties of the hydrogels is absent. 

The paper must be carefully checked for grammatical mistakes and typing errors. 

Please add in conclusion information about the future of the thermoresponsive hydrogel systems.

I suggest to cite the following papers:

https://doi.org/10.1007/s00396-022-04959-1

https://doi.org/10.1007/s00396-020-04750-0

https://doi.org/10.1039/D0SM01505A

Author Response

Reply to reviewers’ comments to:

Ms. Ref. No.: polymers-1748682

Title: Functional Thermoresponsive Hydrogel Molecule to Material Design for Biomedical Applications

The authors are thankful to each reviewer for their thorough investigation of the manuscript and the thoughtful comments on the same. The constructive suggestions by the reviewers have certainly improved the understanding of the manuscript from its previous state. The authors have addressed all the comments to the best of their knowledge, and any changes arising from them have been made in the manuscript and highlighted in colors for the reviewers. The changes are also categorically pointed out in the replies to the comments, which are as follows:

Reviewer #2

The review titled as "Functional Thermoresponsive Hydrogel Molecule to Material Design for Biomedical Applications" summarize materials about thermoresponsive hydrogel for biomedical applications. The review can be published in Polymers MDPI after major revision. 

The following questions may be clarified.

In the review is absent information about nanogels and hydrogel coatings.

Response: Thank you for the comments as suggested by the reviewer, the details of nanogel and hydrogel applications have been included in the introduction part of the revised manuscript.

Please add chemical structures which were mentioned in the review.

Response: Thank you for the comments as suggested by the reviewer, the chemical structures were added in the revised manuscript. 

Information about different gels systems should be summarized in Table.

Response: As suggested the recently published investigations related to hydrogels were summarized in Table 4 and included in the revised manuscript.

What means "In radical polymerization, most monomers include hydrophilic –C C– groups [45]." (line 322-333) It is not clear to me.

Response: The sentence has been corrected in the revised version of the manuscript.

In general, subsection 4. "Mechanism of functional thermoresponsive hydrogel" is difficult for understanding and should be revised. Transition for the PNIPAM was described incorrectly (what about the interaction between amide fragments at T above LCST). Information about a phenomenon named as hydrophobic hydration is absent.

Response: Thank you for the comments, as per suggestion, the updated text is included under the headline of phase transition mechanism.

Information about the impact of the buffer solution or body fluid on the temperature-responsive properties of the hydrogels is absent. 

Response: The suggested part is included under the heading of “applications in drug delivery” in the revised version of the manuscript.

The paper must be carefully checked for grammatical mistakes and typing errors. 

      Response: Thank you for the comments. As suggested by the reviewer we have corrected the grammar and rectified the incomplete sentences in the final version of the manuscript.

Please add in conclusion information about the future of the thermoresponsive hydrogel systems.

Response: Thank you for the comments, as suggested, the future scope of the hydrogel system included in the revised version of the manuscript.

I suggest to cite the following papers:

https://doi.org/10.1007/s00396-022-04959-1

https://doi.org/10.1007/s00396-020-04750-0

https://doi.org/10.1039/D0SM01505A

Response: Thank you for the comments, the suggested papers have been included in the revised version of the manuscript.

Reviewer 3 Report

The article entitled “Functional Thermoresponsive Hydrogel Molecule to Material Design for Biomedical Applications” is a preparatory review useful to know advantages of hydrogels, their preparation, properties and use in biomedical applications, in particular for drug delivery.

In the first section of the article, stimuli-responsive polymers and experimental trials are presented, in order to have a general overview about the argument. A brief list of all kinds of preparation methods is also presented.

In the second section, a deep understanding of thermoresponsive hydrogels is given. The distinction between LCST and UCST, their functional mechanisms and phase transition are studied. 

In the third section various applications of hydrogels are provided, with particular attention to drug delivery, about which also a mathematical description of realise mechanisms is given. A list of commercial products is also presented.

The review provides an overview of thermoresponsive hydrogels in all aspects, from polymers that can be chosen for their preparation to their applications. All the sections are studied in deep in all their parts. The presentation of mathematical models for drug release from the hydrogel-based formulations is very clear and useful to better understand the functional mechanism of hydrogels in drug delivery, one of their most used applications. Tables are well organised and clear.

Some revisions are needed:

-       Check in the text some typing errors,

-       Check also some grammar errors, in many cases sentences are not complete and not very clear,

-       Since the focus of this review are thermoresponsive hydrogels, it would be better to give them particular attention in every section and to underline for each application the utility to use them, because in some cases only advantages of other kinds of stimuli-responsive hydrogels are provided. For example, a deep, clear and useful analysis of drug delivery is presented in paragraph 5 and in subparagraph 6.2, but little information is given about them.

Author Response

Reply to reviewers’ comments to:

Ms. Ref. No.: polymers-1748682

Title: Functional Thermoresponsive Hydrogel Molecule to Material Design for Biomedical Applications

The authors are thankful to each reviewer for their thorough investigation of the manuscript and the thoughtful comments on the same. The constructive suggestions by the reviewers have certainly improved the understanding of the manuscript from its previous state. The authors have addressed all the comments to the best of their knowledge, and any changes arising from them have been made in the manuscript and highlighted in colors for the reviewers. The changes are also categorically pointed out in the replies to the comments, which are as follows:

Reviewer #3

The article entitled “Functional Thermoresponsive Hydrogel Molecule to Material Design for Biomedical Applications” is a preparatory review useful to know advantages of hydrogels, their preparation, properties and use in biomedical applications, in particular for drug delivery.

In the first section of the article, stimuli-responsive polymers and experimental trials are presented, in order to have a general overview about the argument. A brief list of all kinds of preparation methods is also presented.

In the second section, a deep understanding of thermoresponsive hydrogels is given. The distinction between LCST and UCST, their functional mechanisms and phase transition are studied. 

In the third section various applications of hydrogels are provided, with particular attention to drug delivery, about which also a mathematical description of realise mechanisms is given. A list of commercial products is also presented.

The review provides an overview of thermoresponsive hydrogels in all aspects, from polymers that can be chosen for their preparation to their applications. All the sections are studied in deep in all their parts. The presentation of mathematical models for drug release from the hydrogel-based formulations is very clear and useful to better understand the functional mechanism of hydrogels in drug delivery, one of their most used applications. Tables are well organised and clear.

Some revisions are needed:

-  Check in the text some typing errors,

Response: Thank you for the comments, we have thoroughly checked the text and typing error, it was corrected in the final version of the manuscript.

-  Check also some grammar errors, in many cases sentences are not complete and not very clear,

Response: Thank you for the comments. As suggested by the reviewer we have corrected the grammar and rectified the incomplete sentences in the final version of the manuscript.

-    Since the focus of this review are thermoresponsive hydrogels, it would be better to give them particular attention in every section and to underline for each application the utility to use them, because in some cases only advantages of other kinds of stimuli-responsive hydrogels are provided. For example, a deep, clear and useful analysis of drug delivery is presented in paragraph 5 and in subparagraph 6.2, but little information is given about them.

Response: Thank you for the comments, as per suggestion, detailed information is included with respect to effect of biological fluid on the hydrogel characteristics and their impact on drug delivery. Moreover, recently published work was summarized in Table 4 and included in the revised version of the manuscript.

Round 2

Reviewer 2 Report

After revision, the quality of the manuscript was essentially improved. The numerous typing errors can be corrected during the stage of the English correction. I would like to recommend to accept the papers in present form.